# A spike sorting toolbox for up to thousands of electrodes validated with ground truth recordings in vitro and in vivo

Pierre Yger[1†], Giulia LB Spampinato[1†], Elric Esposito[1†], Baptiste Lefebvre[1], Stéphane Deny[1], Christophe Gardella[1,2], Marcel Stimberg[1], Florian Jetter[3], Guenther Zeck[3], Serge Picaud[1], Jens Duebel[1], Olivier Marre[1*]

[1]Institut de la Vision, INSERM UMRS 968, UPMC UM 80, Paris, France; [2]Laboratoire de Physique Statistique, CNRS, ENS, UPMC, 75005, Paris, France; [3]NMI, Neurophysics Group, Reutlingen, Germany

**Abstract** In recent years, multielectrode arrays and large silicon probes have been developed to record simultaneously between hundreds and thousands of electrodes packed with a high density. However, they require novel methods to extract the spiking activity of large ensembles of neurons. Here, we developed a new toolbox to sort spikes from these large-scale extracellular data. To validate our method, we performed simultaneous extracellular and loose patch recordings in rodents to obtain 'ground truth' data, where the solution to this sorting problem is known for one cell. The performance of our algorithm was always close to the best expected performance, over a broad range of signal-to-noise ratios, in vitro and in vivo. The algorithm is entirely parallelized and has been successfully tested on recordings with up to 4225 electrodes. Our toolbox thus offers a generic solution to sort accurately spikes for up to thousands of electrodes.
DOI: https://doi.org/10.7554/eLife.34518.001

*For correspondence:
olivier.marre@gmail.com

†These authors contributed equally to this work

Competing interests: The authors declare that no competing interests exist.

## Introduction

As local circuits represent information using large populations of neurons throughout the brain (*Buzsáki, 2010*), technologies to record hundreds or thousands of them are therefore essential. One of the most powerful and widespread techniques for neuronal population recording is extracellular electrophysiology. Recently, newly developed microelectrode arrays (MEA) have allowed recording of local voltage from hundreds to thousands of extracellular sites separated only by tens of microns (*Berdondini et al., 2005*; *Fiscella et al., 2012*; *Lambacher et al., 2004*), giving indirect access to large neural ensembles with a high spatial resolution. Thanks to this resolution, the spikes from a single neuron will be detected on several electrodes and produce extracellular waveforms with a characteristic spatio-temporal profile across the recording sites. However, this high resolution comes at a cost: each electrode receives the activity from many neurons. To access the spiking activity of individual neurons, one needs to separate the waveform produced by each neuron and identify when it appears in the recording. This process, called spike sorting, has received a lot of attention for recordings with a small number of electrodes (typically, a few tens of electrodes). However, for large-scale and dense recordings, it is still unclear how to extract the spike contributions from extracellular recordings. In particular, for thousands of electrodes, this problem is still largely unresolved.

Classical spike sorting algorithms cannot process this new type of data for several reasons. First, many algorithms do not take into account that the spikes of a single neuron will evoke a voltage deflection on many electrodes. Second, most algorithms do not scale up to hundreds or thousands

of electrodes in vitro and in vivo, because their computation time would increase exponentially with the number of electrodes (*Rossant et al., 2016*). A few algorithms have been designed to process large-scale recordings (*Marre et al., 2012*; *Pillow et al., 2013*; *Pachitariu et al., 2016*; *Leibig et al., 2016*; *Hilgen et al., 2017*; *Chung et al., 2017*; *Jun et al., 2017*), but they have not been tested on real 'ground truth' data.

In ground truth data, one neuron is cherry picked from among all the neurons recorded using an extracellular array using another technique, and simultaneousy recorded. Unfortunately, such data are rare. Dual loose patch and extracellular recordings have been performed for culture of neurons or in cortical slices (*Anastassiou et al., 2015*; *Franke et al., 2015*). However, in this condition, only one or two neurons emit spikes, and this simplifies drastically the spike sorting problem. Ground truth data recorded in vivo are scarce (*Henze et al., 2000*; *Neto et al., 2016*) and in many cases the patch-recorded neuron is relatively far from the extracellular electrodes. As a result, most algorithms have been tested in simulated cases where an artificial template is added at random times to an actual recording. However, it is not clear if this simulated data reproduce the conditions of actual recordings. In particular, waveforms triggered by a given neuron can vary in amplitude and shape, and most simulations do not reproduce this feature of biological data. Also, spike trains of different cells are usually correlated, and these correlations can make extracellular spikes that do overlap.

Here, we present a novel toolbox for spike sorting in vitro and in vivo, validated on ground truth recordings. Our sorting algorithm is based on a combination of density-based clustering and template matching. To validate our method, we performed experiments where a large-scale extracellular recording was performed while one of the neurons was recorded with a patch electrode. We showed that the performance of our algorithm was always close to an optimal classifier, both in vitro and in vivo. We demonstrate that our sorting algorithm could process recordings from up to thousands of electrodes with similar accuracy. To handle data from thousands of electrodes, we developed a tool automating the step that is usually left to manual curation.

Our method is a fast and accurate solution for spike sorting for up to thousands of electrodes, and we provide it as a freely available software that can be run on multiple platforms and several computers in parallel. Our ground truth data are also publicly available and will be a useful resource to benchmark future improvements in spike sorting methods.

## Results

### Spike sorting algorithm

We developed an algorithm (called SpyKING CIRCUS) with two main steps: a clustering followed by a template matching step (see Materials and methods for details). First, spikes are detected as threshold crossings (*Figure 1A*) and the algorithm isolated the extracellular waveforms for a number of randomly chosen spike times. In the following text, we will refer to the extracellular waveforms associated with a given spike time as snippets.

We divided the snippets into groups, depending on their physical positions: for every electrode we grouped together all the spikes having their maximum peak on this electrode. Thanks to this division, the ensemble of spikes was divided into as many groups as there were electrodes. The group associated with electrode $k$ contains all the snippets with a maximum peak on electrode $k$. It was possible that, even among the spikes peaking on the same electrode, there could be several neurons. We thus performed a clustering separately on each group, in order to separate the different neurons present in a single group.

For each group, the snippets were first masked: we assumed that a single cell can only influence the electrodes in its vicinity, and only kept the signal on electrodes close enough to the peak (*Figure 1B*, see Materials and methods). Due to to this reduction, the memory needed for each clustering did no scale with the total number of electrodes. The snippets were then projected into a lower dimensional feature space using Principal Component Analysis (PCA) (usually five dimensions, see Materials and methods), as is classically done in many spike sorting algorithms (*Rossant et al., 2016*; *Einevoll et al., 2012*). Note that the simple division in groups before clustering allowed us to parallelize the clustering step, making it scalable for even thousands of electrodes. ierreEven if a spike is detected on several electrodes, it is only assigned to the electrode where the voltage peak

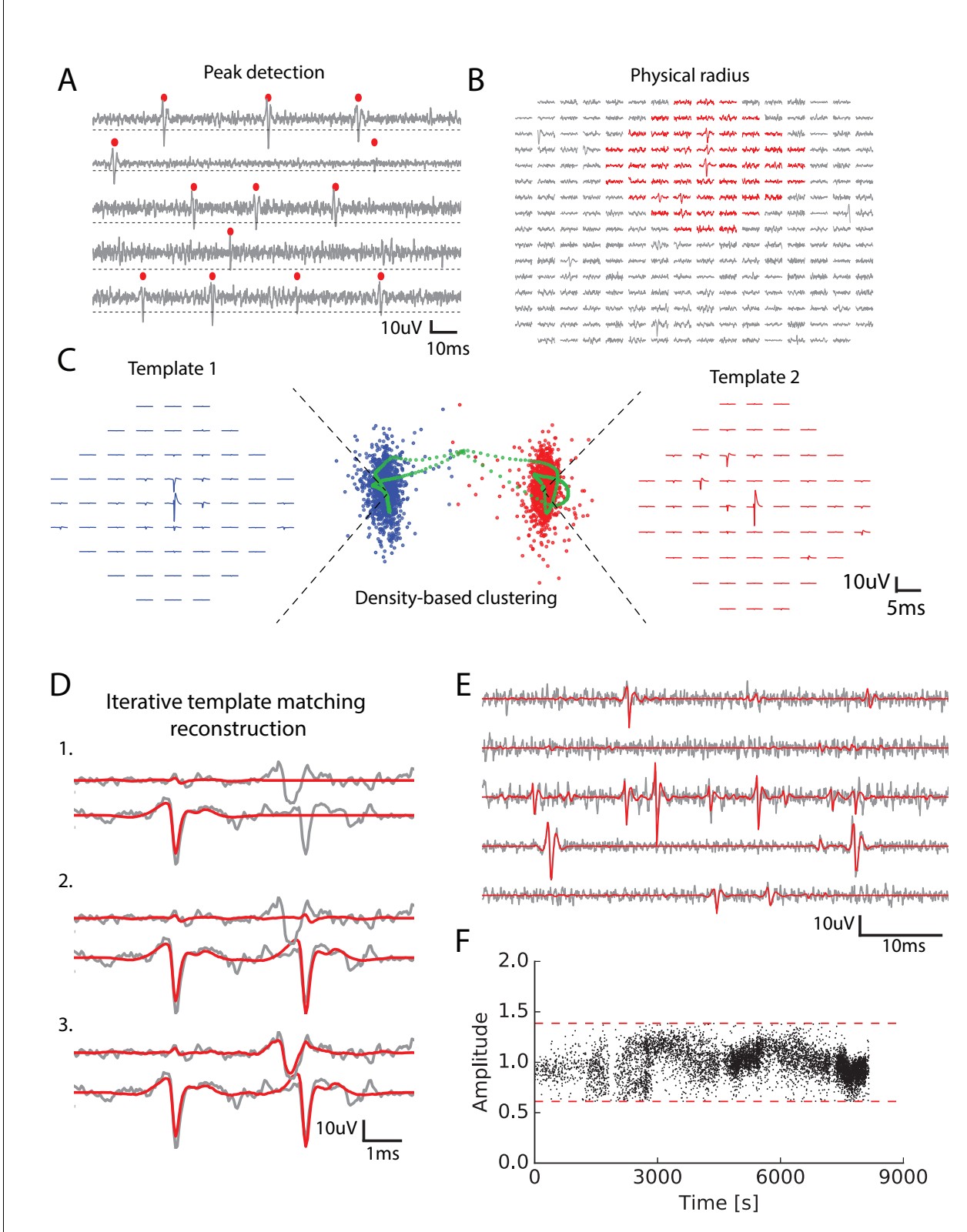

**Figure 1.** Main steps of the spike sorting algorithm. (**A**) Five randomly chosen electrodes, each of them with its own detection threshold (dash dotted line). Detected spikes, as threshold crossings, are indicated with red markers (**B**) Example of a spike in the raw data. Red: electrodes that can be affected by the spike, that is the ones close enough to the electrode where the voltage peak is the highest; gray: other electrodes that should not be affected. (**C**) Example of two clusters (red and blue)with associated templates. Green points show possible combinations of two overlapping spikes

*Figure 1 continued on next page*

*Figure 1 continued*

from the two cells for various time delays. (D) Graphical illustration of the template matching for in vitro data (see Materials and methods). Every line is a electrode. Grey: real data. Red: sum of the templates added by the template matching algorithm; top to bottom: successive steps of the template matching algorithm. E. Final result of the template matching. Same legend as (D, F) Examples of the fitted amplitudes for the first component of a given template as a function of time. Each dot correspond to a spike time at which this particular template was fitted to the data. Dashed dotted lines represent the amplitude thresholds (see Materials and methods).

DOI: https://doi.org/10.7554/eLife.34518.002

The following figure supplement is available for figure 1:

**Figure supplement 1.** Schematic of the parallel clustering of the spikes, in a toy example with two electrodes.

DOI: https://doi.org/10.7554/eLife.34518.003

is the largest: if a spike has its largest peak on electrode 1, but is also detected on electrode 2, it will only be assigned to electrode1 (see *Figure 1—figure supplement 1*).

For each group, we performed a density-based clustering inspired by (*Rodriguez and Laio, 2014*) (see Materials and methods). The idea of this algorithm is that the centroid of a cluster in the feature space should have many neighbours, that is a high density of points in their neighborhood. The centroid should also be the point where this density is a local maximum: there should not be a point nearby with a higher density. To formalize this intuition, for each point we measured the average distance of the 100 closest points $rho$ (intuitively, this is inversely proportional to the local density of points), and the distance $\delta$ to the closest point of higher density (i.e. with a lower $\rho$). Centroids should have a low $\rho$ and a high $\delta$. We hypothesized that there was a maximum of ten clusters in each group (i.e. at most ten different cells peaking on the same electrode) and took the ten points with the largest $\delta/\rho$ ratio as the centroids. Each remaining point was then assigned iteratively to the nearest point with highest density, until they were all assigned to the centroids (see Materials and methods for details - note that all the numbers mentioned here are parameters that are tunable in our toolbox).

Thanks to this method we could find many clusters, corresponding to putative neurons. In many spike-sorting methods, it is assumed that finding clusters is enough to solve the spike-sorting problem (*Chung et al., 2017*). However, this neglects the specific issue of overlapping spikes (see *Figure 1C*). When two nearby cells spike synchronously, the extracellular waveforms evoked by each cell will superimpose (*Figure 1C*, see also [*Pillow et al., 2013*]). This superimposition of two signals coming from two different cells will distort the feature estimation. As a result, these spikes will appear as points very far away from the cluster associated to each cell. An example of this phenomena is illustrated in *Figure 1C*. Blue and red points correspond to the spikes associated to two different cells. In green, we show the spikes of one cell when the waveform of another one was added at different delays. For short delays, the presence of this additional waveform strongly distort the feature estimation. As a result, the corresponding point is far from the initial cluster, and will be missed by the clustering. To overcome this issue, we performed a template matching as the next step of our algorithm.

For this, we first extracted a template from each cluster. This template is a simplified description of the cluster and is composed of two waveforms. The first one is the average extracellular waveform evoked by the putative cell (*Figure 1C*, left and red waveforms). The second is the direction of largest variance that is orthogonal to this average waveform (see Materials and methods). We assumed that each waveform triggered by this cell is a linear combination of these two components. Thanks to these two components, the waveform of the spikes attributed to one cell could vary both in amplitude and in shape.

At the end of this step, we should have extracted an ensemble of templates (i.e. pairs of waveforms) that correspond to putative cells. Note that we only used the clusters to extract the templates. Our algorithm is thus tolerant to some errors in the clustering. For example, it can tolerate errors in the delineation of the cluster border. The clustering task here is therefore less demanding than in classical spike sorting algorithms where finding the correct cluster borders is essential to minimize the final error rate. By focusing on only getting the cluster centroids, we should thus have made the clustering task easier, but all the the spikes corresponding to one neuron have yet to be found. We therefore used a template matching algorithm to find all the instances where each cell has emitted a spike.

In this step, we assumed that the templates of different cells spiking together sum linearly and used a greedy iterative approach inspired by the projection pursuit algorithm to match the templates to the raw data (*Figure 1D*, see Materials and methods). Within a piece of raw data, we looked for the template whose first component had the highest similarity to the raw signal (here similarity is defined as the scalar product between the first component of the template and the raw data) and matched its amplitude to the signal. If this amplitude falls between pre-determined thresholds (see Materials and methods), we matched and subtracted the two components to the raw signal. These predetermined thresholds reflect the prior that the amplitude of the first component should be around 1, which corresponds to the average waveform evoked by the cell. We then re-iterated this matching process until no more spike could be matched (*Figure 1D,E*) (see Materials and methods). We found many examples where allowing amplitude variation wasa desirable feature (see *Figure 1F*).

After this template matching step, the algorithm outputs putative cells, described by the templates, and associated spike trains, that is spike times where the template was matched to the data.

## Performance on ground truth data

To test our algorithm, we performed dual recordings (*Figure 2A,B*) using both a multielectrode array to record many cells (see schematic on *Figure 2A*), and simultaneous loose patch to record the spikes of one of the cell (*Figure 2B*). For this cell we know what should be the output of the spike sorting. In vitro, we recorded 18 neurons from 14 retinas with a 252 electrode MEA where the spacing between electrodes was 30 $\mu$m (see Materials and methods, (*Spampinato et al., 2018*)). We also generated datasets where we removed the signals of some electrodes, such that the density of the remaining electrodes was either 42 or 60 $\mu$m (by removing half or 3 quarters of the electrodes, respectively).

We then ran the spike sorting algorithm only on the extracellular data, and estimated the error rate (as the mean of false positives and false negatives, see Materials and methods) for the cell recorded in loose patch, where we know where the spikes occurred. The performance of the algorithm is limited not only by imperfections of the algorithm, but also by several factors related to the ground truth data themselves. In particular, some of the cells recorded with loose patch can evoke only a small spike on the extracellular electrode, for example if they are far from the nearest electrode or if their soma is small (*Buzsáki, 2004*). If a spike of the patch-recorded cell triggers a large voltage deflection, this cell should be easy to detect. However, if the triggered voltage deflection is barely detectable, even the best sorting algorithm will not perform well. Looking at *Figure 2C*, for in vitro data (see Materials and methods), we found that there was a correlation between the error rate of our algorithm and the size of the extracellular waveform evoked by the spikes of the patch-recorded cell: the higher the waveform, the lower the error rate.

We then asked if our algorithm is close to the 'best' possible performance, that is the only errors are due to intrinsic limitations in the ground truth data (e.g. small waveform size).

There is no method to exactly estimate this best possible performance. However, a proxy can be found by training a nonlinear classifier on the ground truth data (*Harris et al., 2000*; *Rossant et al., 2016*). We trained a nonlinear classifier on the extracellular waveforms triggered by the spikes of the recorded cell, similar to (*Harris et al., 2000*; *Rossant et al., 2016*) (referred to as the Best Ellipsoidal Error Rate (BEER), see Materials and methods). This classifier 'knows' where the true spikes are and simply quantifies how well they can be separated from the other spikes based on the extracellular recording. Note that, strictly speaking, this BEER estimate is not a lower bound of the error rate. It assumes that spikes can be all found inside a region of the feature space delineated by ellipsoidal boundaries. As we have explained above, spikes that overlap with spikes from another cell will probably be missed and this ellipsoidal assumption is also likely to be wrong in case of bursting neurons or electrode-tissue drifts. However, we used the BEER estimate because it has been used in several papers describing spike sorting methods (*Harris et al., 2000*; *Rossant et al., 2016*) and has been established as a commonly accepted benchmark. In addition, because we used rather stationary recordings (few minutes long, see Materials and methods), we did not see strong electrode-tissue drifts.

We estimated the error made by the classifier and found that the performance of our algorithm almost always was in the same order of magnitude as the performance of this classifier, (*Figure 2D*, left; $r = 0.89$, $p < 10^{-5}$) over a broad range of spike sizes. For 37 neurons with large waveform sizes

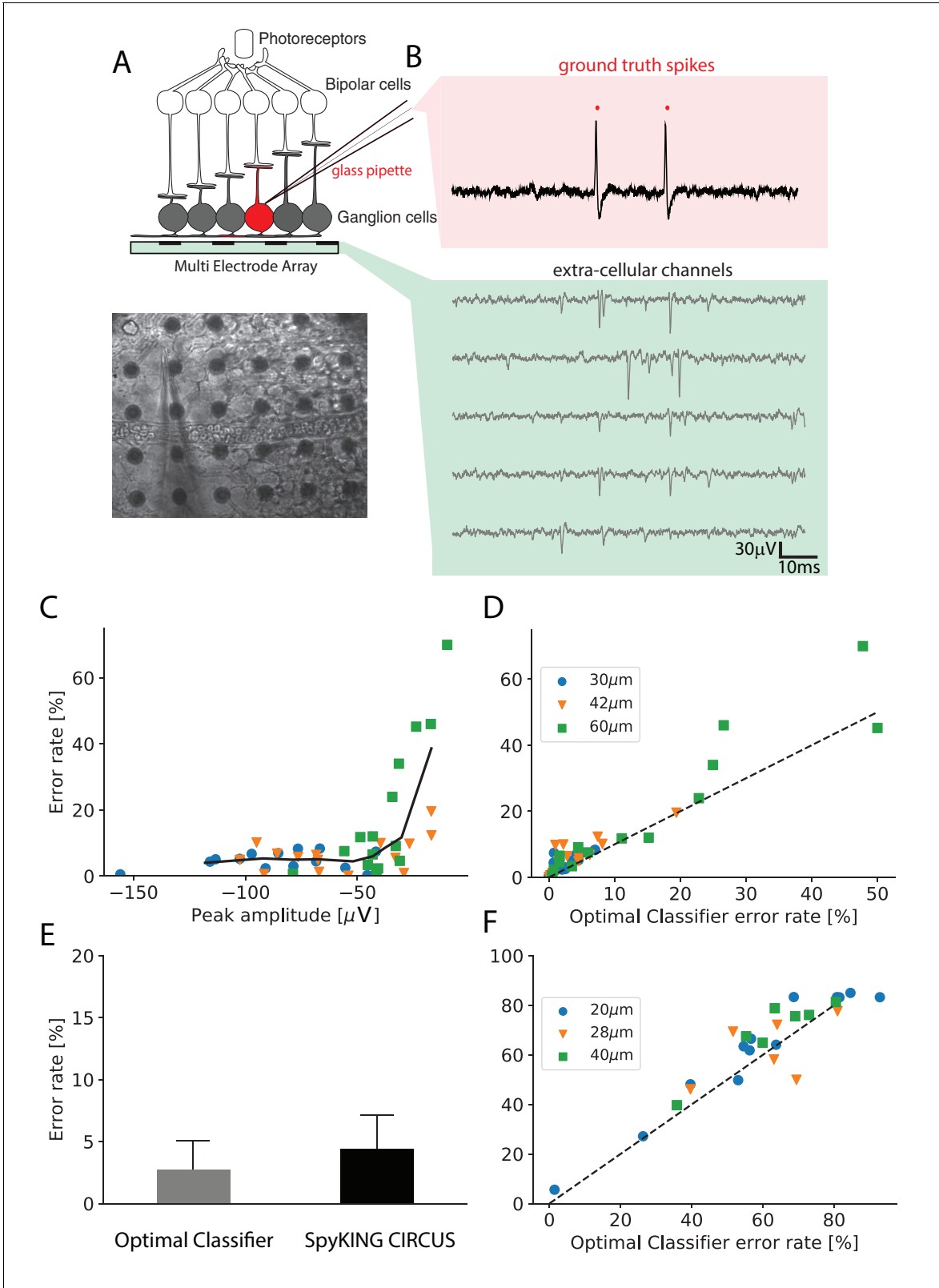

**Figure 2.** Performance of the algorithm on ground truth datasets. (**A**) Top: Schematic of the experimental protocol in vitro. A neuron close to the multielectrode array (MEA) recording is recorded in loose patch. Bottom: Image of the patch electrode on top of a 252 electrodes MEA, recording a ganglion cell. (**B**) Top, pink box: loose patch recording showing the spikes of the recorded neuron. Bottom, green box: Extra-cellular recordings next to the loose patched soma. Each line is a different electrode (**C**) Error rate of the algorithm as function of the largest peak amplitude of the ground-truth

*Figure 2 continued on next page*

*Figure 2 continued*

neuron, recorded extracellularly in vitro. (D) Comparison between the error rates produced by the algorithm on different ground truth datasets and the error rates of nonlinear classifiers (Best Ellipsoidal Error Rate) trained to detect the spikes for in vitro data (*Spampinato et al., 2018*). (E) Comparison of average performance for all neurons detected by the Optimal Classifier with an error less than 10% (n = 37). F. Same as D. but for in vivo data (*Neto et al., 2016*) (see Materials and methods).

DOI: https://doi.org/10.7554/eLife.34518.004

(above) the average error of the classifier is 2.7% and the one for our algorithm is 4.8% (see *Figure 2E*). For 22 neurons with lower spike size (below), the average error of the classifier is 11.1% and the one for our algorithm is 15.2%. This suggests that our algorithm reached an almost optimal performance on this in vitro data.

We also used similar ground truth datasets recorded in vivo in rat cortex using dense silicon probes with either 32 or 128 recording sites (*Neto et al., 2016*). With the same approach as for in vitro data, we also found that our algorithm achieved near optimal performance (*Figure 2F*, right; $r = 0.92$, $p < 10^{-5}$), although there were only two recordings where the spike size of the patch-recorded neuron was large enough to be sorted with a good accuracy. For only two available neurons with low optimal error rate, the average error of the classifier is 13.9 and the one for our algorithm is 14.8%. For 24 neurons with lower spike size, the average error of the classifier is 64.0% and the one for our algorithm is 67.8%. Together, these results show that our algorithm can reach a satisfying performance (i.e. comparing to the classifier error) over a broad range of spike sizes, for both in vivo and in vitro recordings.

We also compared our performance to the Kilosort algorithm (*Pachitariu et al., 2016*) and found similar performances (4.4% on average over all non-decimated neurons for SpyKING CIRCUS against 4.2% for Kilosort). Because Kilosort used a GPU, it could be run faster than our algorithm on a single machine: on a 1 hr recording with 252 electrodes, Kilosort can achieve a four times speedup on a standard desktop machine (see Materials and methods). But without using a GPU, Kilosort was only marginally faster (1.5 speedup), and slower if we started using several cores of the machine. However, is it worth noticing that the speedup of Kilosort comes at the cost of an increased usage of memory. Kilosort used 32 GB of RAM for a maximal number of 500 neurons, while our algorithm had a much lower memory footprint, because of design choices. We have therefore found a trade off where execution speed is slightly slower, but much less memory is used. Thanks to this, we could run our algorithm to process recordings with thousands of electrodes, while Kilosort does not scale up to this number. In the next section, we demonstrate that our algorithm still works accurately at that scale.

## Scaling up to thousands of electrodes

A crucial condition to process recordings performed with thousands of electrodes is that every step of the algorithm should be run in parallel over different CPUs. The clustering step of our algorithm was run in parallel on different subsets of snippets as explained above. The template matching step could be run independently on different blocks of data, such that the computing time only scaled linearly with the data length. Each step of the spike sorting algorithm was therefore parallelized. The runtime of the full algorithm decreased proportionally with the numbers of CPU cores available (*Figure 3A*, grey area indicates where the software is 'real-time' or faster). As a result, the sorting algorithm could process 1 hr of data recorded with 252 electrodes in 1 hr with 9 CPU cores (spread over three computers) (*Figure 3A,B*). It also scaled up linearly with the number of electrodes (*Figure 3B*), and with the number of templates (*Figure 3C*). It was therefore possible to run it on long recordings with more than 4000 electrodes, and the runtime could be be divided by simply having more CPUs available.

To test the accuracy of our algorithm on 4225 electrodes, we generated hybrid ground truth datasets where artificial spikes were added to real recordings performed on mouse retina in vitro (see Materials and methods). We ran the spike-sorting algorithm on different datasets, picked some templates and used them to create new artificial templates that we added at random places to the real recordings (see Materials and methods). This process, as shown in *Figure 3D* allowed us to generate 'hybrid' datasets were we know the activity of a number of artificially injected neurons. We

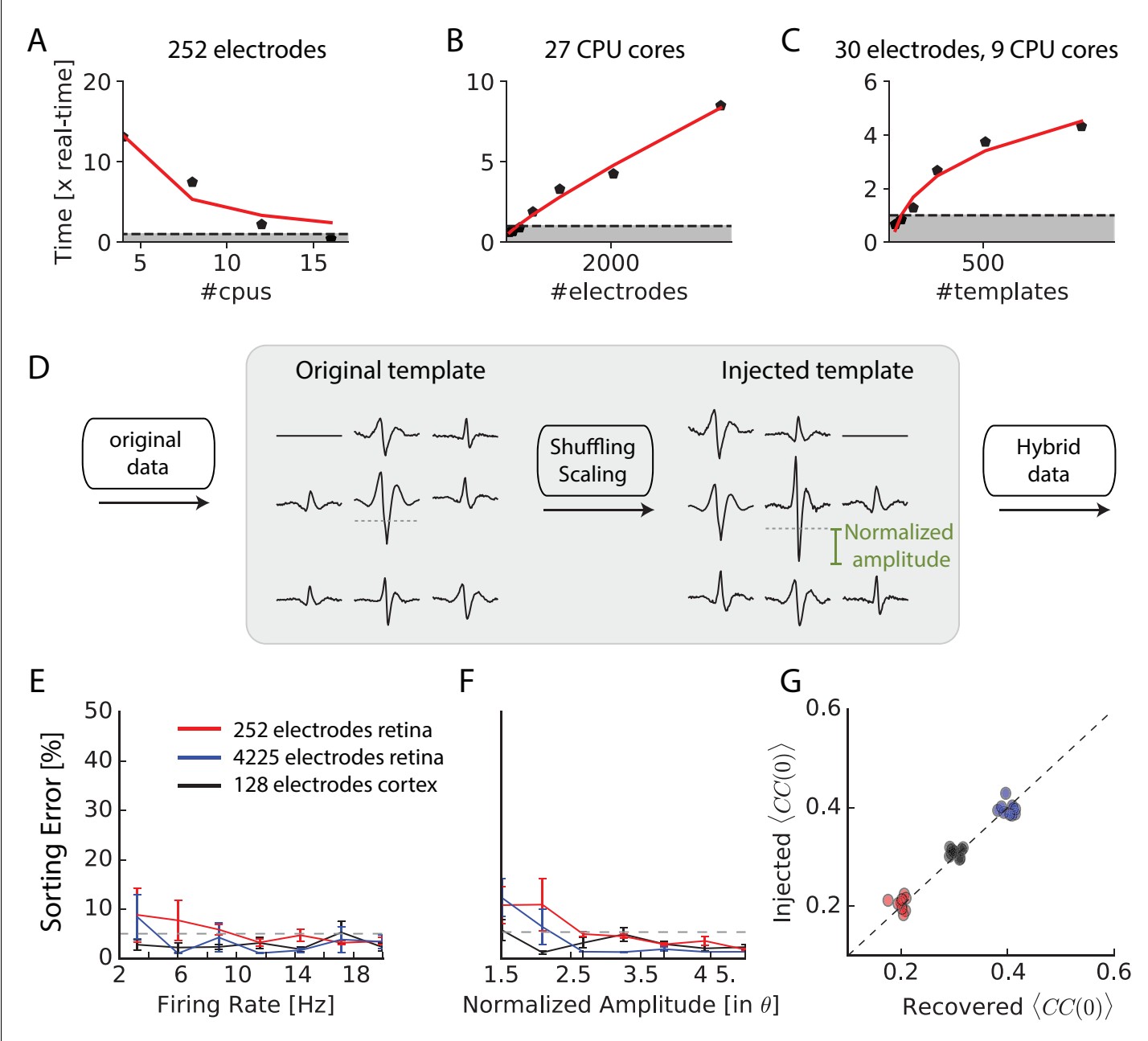

**Figure 3.** Scaling to thousands of electrodes. (A) Execution time as function of the number of processors for a 90 min dataset in vitro with 252 electrodes, expressed as a real-time ratio, that is the number of hours necessary to process one hour of data. (B) Execution time as function of the number of electrodes for a 30 min dataset recorded in vitro with 4225 electrodes. (C) Execution time as function of the number of templates for a 10-min synthetic dataset with 30 electrodes. (D) Creation of 'hybrid' datasets: chosen templates are injected elsewhere in the data as new templates. Dashed-dotted lines shows the detection threshold on the main electrode for a given template, and normalized amplitude is expressed relative to this threshold (see Materials and methods). (E) Mean error rate as function of the firing rate of injected templates, in various datasets. Errors bars show standard error over eight templates (F) Error rate as function of the normalized amplitude of injected templates, in various datasets. Errors bars show standard error over nine different templates (G) Injected and recovered cross-correlation value between pairs of neurons for five templates injected at 10 Hz, with a normalized amplitude of 2 (see Materials and methods).

DOI: https://doi.org/10.7554/eLife.34518.005

then ran our sorting algorithm on these datasets and measured if the algorithm was able to find at which times the artificial spikes were added. We counted a false-negative error when an artificial spike was missed and a false-positive error when the algorithm detected a spike when there was not any (see Materials and methods). Summing these two types of errors, the total error rate remained below 5% for all the spikes whose size was significantly above spike detection threshold (normalized amplitude corresponds to the spike size divided by the spike threshold). Error rates were similar for recordings with 4255 electrodes in vitro, 128 electrodes in vivo or 252 electrodes in vitro. Performance did not depend on the firing rate of the injected templates (*Figure 3E*) and only weakly on the normalized amplitude of the templates (*Figure 3F*), as long as it was above the spike threshold. The accuracy of the algorithm is therefore invariant to the size of the recordings.

A crucial issue when recording thousands of densely packed electrodes is that more and more spikes overlap with each other. If an algorithm misses overlapping spikes, then the estimation of the amplitude of correlations between cells will be biased. To test if our method was able to solve the problem of overlapping spikes and thus estimate correlations properly, we generated hybrid datasets where we injected templates with a controlled amount of overlapping spikes (see Materials and methods). We then ran the sorting algorithm and compared the injected spike trains and the spike trains recovered by SpyKING CIRCUS. We then compared the correlation between both pairs. If some overlapping spikes were missed by the algorithm, the correlation between the sorted spike trains should be lower than the correlation between the injected spike trains. We found that our method was always able to estimate the pairwise correlation between the spike trains with no under-estimation (*Figure 3G*). Overlapping spikes were therefore correctly detected by our algorithm. The ability of our template matching algorithm to resolve overlapping spikes thus allowed an unbiased estimation of correlations between spike trains, even for thousands of electrodes.

These different tests, described above, show that SpyKING CIRCUS reached a similar performance for 4225 electrodes than for hundreds electrodes, where our ground truth recordings showed that performance was near optimal. Our algorithm is therefore also able to sort accurately recordings from thousands of electrodes.

## Automated merging

As in most spike-sorting algorithms, our algorithm may split one cell into several units. After running the entire algorithm, it is therefore necessary to merge together the units corresponding to the same cell. However, for hundreds or thousands of electrodes, going through all the pairs of units and merging them by hand would take a substantial amount of time. To overcome this problem, we designed a tool to merge automatically many units at once so that the time spent on this task does not scale with the number of electrodes and this allows us to automate this final step.

Units that likely belong to the same cell (and thus should be merged) have templates that look alike and in addition, the combined cross-correlogram between the two cell's spike trains shows a clear dip near 0 ms, indicating that the merged spike trains do not show any refractory period violation (*Figure 4A*, blue example). In order to automate this merging process, we formalized this intuition by estimating for each pair of units two factors that reflect if they correspond to the same cell or not. For each pair of templates, we estimated first the similarity between templates and second the dip in the center of the cross-correlogram. This dip is measured as the difference between the geometrical mean of the firing rate $\phi$ (i.e. the baseline of the cross-correlogram) and the value of the cross-correlogram at delay 0 ms, $\langle CC \rangle$ (see Materials and methods and right insets in *Figure 4A*).

We plotted for each pair with high similarity the dip estimation against the geometrical mean of their firing rates. If there is a strong dip in the cross-correlogram (quantified by $\phi - \langle CC \rangle$), the dip quantification and the geometrical mean, $\phi$, should be almost equal, and the corresponding pair should thus be close to the diagonal in the plot.

In one example, where we artificially split synthetic spike trains (*Figure 4A*; see Materials and methods), we could clearly isolate a cluster of pairs lying near this diagonal, corresponding to the pairs that needed to be merged (*Figure 4A* right panels). We have designed a GUI such the user can automatically select this cluster and merge all the pairs at once. Thanks to this, with a single manipulation by the user, all the pairs are merged.

We then tested this method on our in vitro ground truth data. In these recordings, the cell recorded with loose patch might be split by the algorithm between different spike trains. We can determine the units that correspond to the patch-recorded cell. For one particular neuron taken

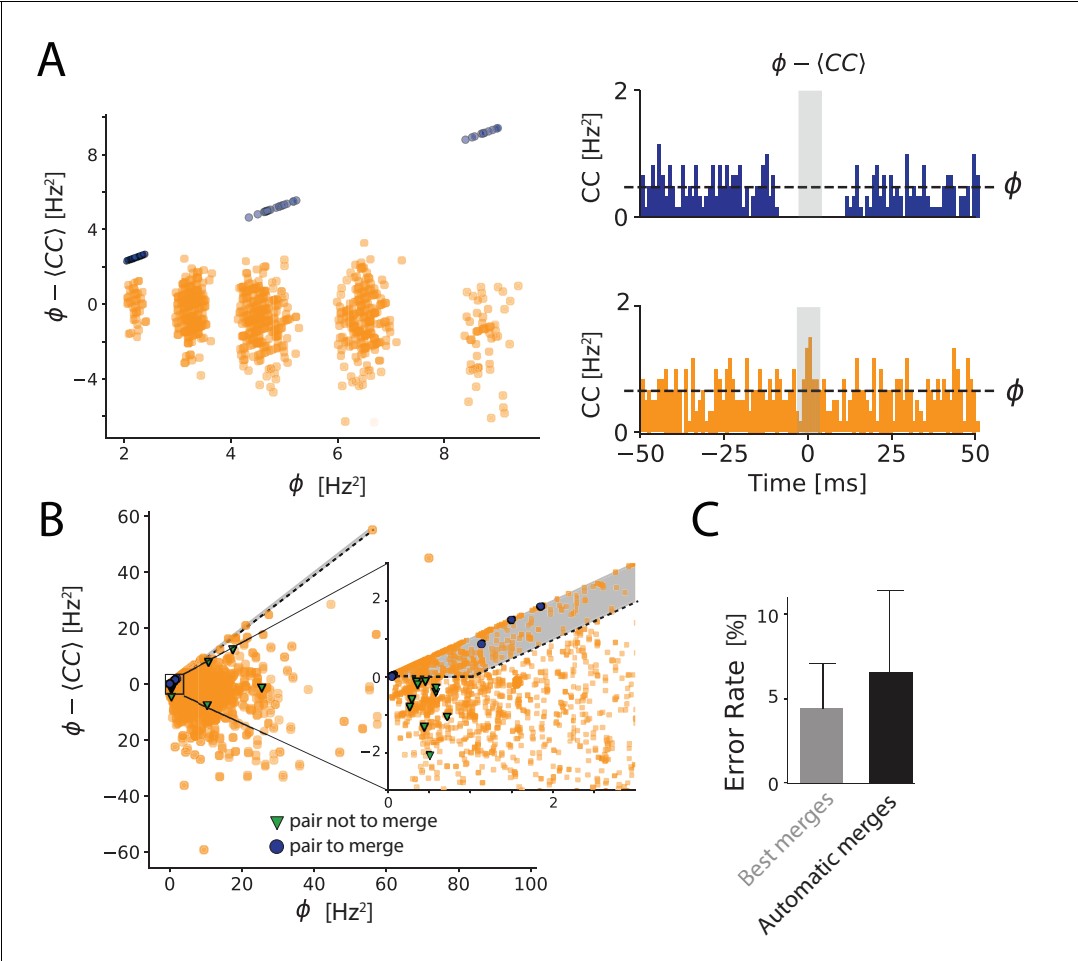

**Figure 4.** Automated merging. (**A**) Dip estimation (y-axis) compared to the geometrical mean of the firing rate (x-axis) for all pairs of units and artificially generated and split spike trains (see Materials and methods). Blue: pairs of templates originating from thesame neuron that have to be merged. Orange: pairs of templates corresponding to different cells. Panels on the right: for two chosen pairs, one that needs to be merged (in blue, top panel) and one should not be merged (orange, bottom panel) the full cross-correlogram and the geometrical mean of the firing rate (dashed line). The average correlation is estimated in the temporal window defined by the gray area. (**B**) Same as A, for a ground truth dataset. Blue points: points that need to be merged. Green points: pairs that should not be merged. Orange points: pairs where our ground truth data does not allow us telling if the pair should be merged or not. The gray area corresponds to the region where pairs are merged by the algorithm. Inset: zoom on one region of the graph. (**C**) Average error rate in the case where the decision of merging units was guided by the ground truth data (left) against the automated strategy designed here (right).

DOI: https://doi.org/10.7554/eLife.34518.006

from our database, we can visualize all the units that need to be merged together (blue points in *Figure 4B*), and that should not be merged with units corresponding to other cells (green pairs in *Figure 4B*). For all the other cells, we do not have access to a ground truth, and thus cannot decide if the pairs should be merged or not (orange pairs in *Figure 4B*).

To automate the merging process entirely, we defined two simple rules to merge two units: first, their similarity should be above a threshold (similarity threshold, 0.8 in our experiments). Second, the dip estimation for this pair should be close to the geometrical mean of firing rates, that is their difference should be below a threshold (dip threshold). In practice, the corresponding point in *Figure 4B* should be above a line parallel to the diagonal. We used these rules to perform a fully automatic merging of all units. We then estimated the error rate for the ground truth cell, in the same way as the previous section. We also estimated the lowest error rate possible error rate by merging the units using the ground truth data for guidance (Best Merges, see aterials and m). We found that the error rate obtained with our automated method was close to this best error rate

(*Figure 4C*). We have therefore automated the process of merging spike trains while keeping a low error rate. The performance did not vary much with the values of the two parameters (similarity threshold and dip threshold), and we used the same parameters for all the different datasets. This shows that the sorting can be fully automated while limiting the error rate to a small value. We thus have a solution to fully automate the sorting, including all the decisions that need to be taken during the manual curation step. However, because we used cross-correlograms in order to help automate the merging process, it is worth noticing that one can no longer use cross-correlograms as a validation metric.

## Discussion

We have shown that our method, based on density-based clustering and template matching, allows sorting spikes from large-scale extracellular recordings both in vitro and in vivo. We tested the performance of our algorithm on 'ground truth' datasets, where one neuron is recorded both with extracellular recordings and with a patch electrode. We showed that our performance was close to an optimal nonlinear classifier, trained using the true spike trains. Our algorithm has also been tested on purely synthetic datasets (*Hagen et al., 2015*) and similar results were obtained (data not shown). Note that tests were performed by different groups on our algorithm and show its high performance on various datasets (see http://spikesortingtest.com/ and http://phy.cortexlab.net/data/sortingComparison/). Our algorithm is entirely parallelized and could therefore handle long datasets recorded with thousands of electrodes. Our code has already been used by other groups (*Denman et al., 2017*; *Mena et al., 2017*; *Chung et al., 2017*; *Wilson et al., 2017*) and is available as a complete multi-platform, open source software for download (http://spyking-circus.rtfd.org) with a complete documentation. Note that all the parameters mentioned in the description of the algorithm can be modified easily to work with different kinds of data. We have made all the ground truth data available for download (see Source Code section in Materials and methods), so that improvements in our algorithm as well as alternative solutions could be benchmarked easily in the future.

Classical approaches to the spike sorting problem involve extracting some features from each detected spike (*Hubel, 1957*; *Meister et al., 1994*; *Lewicki, 1994*; *Einevoll et al., 2012*; *Quiroga et al., 2004*; *Hill et al., 2011*; *Pouzat et al., 2002*; *Litke et al., 2004*; *Chung et al., 2017*) and clustering the spikes in the feature space. In this approach, the spike sorting problem is reduced to a clustering problem and this introduces several major problems. First, to assign the spikes to the correct cell, the different cellsmust be separated in the feature space. Finding the exact borders of each cell in the feature space is a hard task that cannot be easily automated (but see [*Chung et al., 2017*]). Second, running a clustering algorithm on data with thousands of electrodes is very challenging. Finally, overlapping spikes will appear as strong deviations in the feature space and will therefore be missed in this approach. These three issues preclude the use of this approach for large-scale recordings with dense arrays of electrodes. In comparison, here we have parallelized the clustering step efficiently, using a template matching approach, so that we only needed to infer the centroid of each cluster and not their precise borders. The template matching approach also allowed us to deconvolve overlapping spikes in a fast, efficient and automated manner. Some template matching approaches have been previously tested, mostly on in vitro data (*Marre et al., 2012*; *Pillow et al., 2013*; *Franke et al., 2015b*) but were not validated on ground truth datasets like the ones we acquired here. Also, they only had one waveform for each template, which did not allow any variation in the shape of the spike, while we have designed our template matching method to take into account not only variation in the amplitude of the spike waveform, but also in shape. Finally, several solutions did not scale up to thousands of electrodes. All GPU-based algorithms (*Pachitariu et al., 2016*; *Lee et al., 2017*; *Jun et al., 2017*) only scale for a few hundreds channels, and face severe memory issues for larger probes.

Finally, a common issue when sorting spikes from hundreds or thousands of electrodes is the time spent on manual curation of the data. Here, we have designed a tool to automate this step by merging units corresponding to the same cell all at once, based onthe cross-correlogram between cells and the similarity between their templates. Having an objective criterion for merging spike trains not only speeds up the manual curation time, it also makes the results less sensitive to human errors and variability in decisions. In some cases, it might be necessary to take into account

additional variables that are specific to the experiment, but even then our tool will still significantly reduce the time spent on manual curation.

Our method is entirely parallel and can therefore be run in 'real time' (i.e. 1 hr of recording processed in 1 hr) with enough computer power. This paves the way towards online spike sorting for large-scale recordings. Several applications, likebrain machine interfaces, or closed-loop experiments (*Franke et al., 2012*; *Hamilton et al., 2015*; *Benda et al., 2007*), will require an accurate online spike sorting. This will require adapting our method to process data 'on the fly', processing new data blocks when they become available and adapting the shape of the templates over time.

## Materials and methods

### Experimental recordings

#### In vitro recordings with 252 or 4225 electrodes

Retinal tissue was obtained from adult (8 weeks old) male Long-Evans rat (Rattus norvegicus) or mouse (mus musculus, 4–9 weeks old) and continuously perfused with Ames Solution (Sigma-Aldrich) and maintained at 32 C. Ganglion cell spikes were recorded extracellularly from a multielectrode array with 252 electrodes spaced 30 or 60 $\mu$m apart (Multi-Channel Systems) or with 4225 electrodes arranged in a 2D grid and spacedby 16 $\mu$m (*Wilson et al., 2017*),4] at a sampling rate of 20 kHz. Experiments were performed in accordance with institutional animal care standards.

For the ground truth recordings, electrophysiological recordings were obtained from ex vivo isolated retinae of rd1 mice (4/5 weeks old). The retinal tissue was placed in AMES medium (Sigma-Aldrich, St Louis, MO; A1420) bubbled with 95% $O_2$ and 5% $CO_2$ at room temperature, on a MEA (10 $\mu$m electrodes spaced by 30 $\mu$m; Multichannel Systems, Reutlingen, Germany) with ganglion cells layer facing the electrodes. Borosilicate glass electrodes (BF100-50, Sutter instruments) were filled with AMES with a final impedance of 6–9 M$\Omega$. Cells were imaged with a customized inverted DIC microscope (Olympus BX 71) mounted with a high sensitivity CCD Camera (Hamamatsu ORCA −03G) and recorded with an Axon Multiclamp 700B patch clamp amplifier set in current zero mode. We used rd1 mice because going through the photoreceptor layer with the patch pipette was easier than for a wild-type mouse.

For the data shown in *Figures 1* and *3*, we used a recording of 130 min. For the data shown in *Figure 2A*, 16 neurones were recorded over 14 intact retinas. Recording durations all lasted 5 min. The thresholds for the detection of juxta-cellular spikes were manually adjusted for all the recordings (*Spampinato et al., 2018*)

#### In vivo recordings with 128 electrodes

We use the freely available datasets provided by (*Neto et al., 2016*). Those are 32 or 128 dense silicon probes recordings (20 $\mu$m spacing) at 30 kHz performed in rat visual cortex, combined with juxta-cellular recordings. The dataset gave us a total of 13 neurons for *Figure 2*.C with recordings between 4 and 10 min each. Similarly to the in vitro case, the detection thresholds for the juxta-cellular spikes were manually adjusted based on the data provided by *Neto et al. (2016)* and on spike-triggered waveforms. For the validation with 'hybrid' dataset, shown in *Figure 3*, we used the longest dataset recorded with 128 electrodes.

### Details of the algorithm

In the following, we consider that we have $N_{elec}$ electrodes, acquired at a sampling rate $f_{rate}$. Every electrode $k$ is located at a physical position $\mathbf{p}_k = (x_k, y_k)$ in a 2D space (extension to 3D probes would be straightforward). The aims of our algorithm is to decompose the signal as a linear sum of spatio-temporal kernels or 'templates' (see *equation 1*).

$$\mathbf{s}(t) = \sum_{ij} a_{ij}\mathbf{w}_j(t - t_i) + b_{ij}\mathbf{v}_j(t - t_i) + \mathbf{e}(t) \tag{1}$$

where $\mathbf{s}(t)$ is the signal recorded over $N_{elec}$ electrodes and over multiple time points. $\mathbf{w}_j(t - t_i)$ and $\mathbf{v}_j(t - t_i)$ are the two components of the template associated to each cell. They represent the waveform triggered on the electrodes by cell $j$. Times $\{t_i\}$ are the putative spike times over all the electrodes. $a_{ij}$ and $b_{ij}$ are the amplitude factors for spike time $t_i$ for the template $j$, and $\mathbf{e}(t)$ is the

background noise. reNote that at a given spike time $t_i$, it is likely that only a couple of cells fire a spike. Only these cells will therefore have $a_{ij}$ and $b_{ij}$ different from zero. For all the other ones, these coefficients are zero simply because the cell does not fire at this time.

The algorithm can be divided into two main steps, described below. After a preprocessing stage, we first run a clustering algorithm to extract a dictionary of 'templates' from the recording. Second, we use these templates to decompose the signal with a template-matching algorithm. We assume that a spike will only influence the extracellular signal over a time window of size $N_t$ (typically 2 ms for in vivo and 5 ms for in vitro data) and only electrodes whose distance to the soma is below $r_{\max}$ (typically for in vivo and for in vitro data). For every electrode $k$ centered on $\mathbf{p}_k$, we define $G^k$ as the ensemble of nearby electrodes $l$ such that $|\mathbf{p}_k - \mathbf{p}_l|_2 \leq r_{\max}$. The key parameters of the algorithmare summarized in *Table 1*.

## Pre-processing
### Filtering
In a preprocessing stage, all the signals were individually high-pass filtered with a Butterworth filter of order three and a cutoff frequency of to remove any low-frequency components of the signals.We then subtracted, for every electrode $k$, the median such that.

### Spike detection
Once signals have been filtered, we computed a spike threshold $\theta_k$ for every electrode $k$: $\theta_k = \lambda \mathrm{MAD}(\mathbf{s}_k(t))$, where MAD is the Median Absolute Deviation, and $\lambda$ is a free parameter. For all the datasets shown in this paper, we set $\lambda = 6$. We detected the putative spike times $t_i$ as times where there was at least one electrode $k$ where $\mathbf{s}_k(t_i)$ was below the threshold $-\theta_k$ and a local minimum of the voltage $vects_k(t)$.

### Whitening
To remove spurious spatial correlations between nearby recordings electrodes, we performed a spatial whitening on the data. To do so, we searched for a maximum of 20 s of recordings where there were no spikes (i.e no threshold crossings). We then computed the Covariance Matrix of the noise $\mathbf{C}_{\mathrm{spatial}}$ and estimated its eigenvalues $\{d_m\}$ and associated eigenvector matrix $\mathbf{V}$. From the diagonal matrix $\mathbf{D} = \mathrm{diag}\left(\frac{1}{\sqrt{d+\epsilon}}\right)$, where $\epsilon = 10^{-18}$ is a regularization factor to ensure stability, we computed the whitening matrix $\mathbf{F} = \mathbf{VDV^T}$. In the following, each time blocks of data are loaded, they are multiplied by $\mathbf{F}$. After whitening,we recomputed the spike detection threshold $\theta_k$ of each electrode $k$ in the whitened space.

### Basis estimation (PCA)
Our first goal was to reduce the dimensionality of the temporal waveforms. We collected up to $N_\mathrm{p}$ spikes on each electrode. We thus obtained a maximum of $N_\mathrm{p} \times N_\mathrm{elec}$ spikes and took the waveform only on the peaking electrode for each of them. This is a collection of a large number of temporal waveforms and we then aimed at finding the best basis to project them. In order to compensate for sampling rate artifacts, we first upsampled all the collected single-electrode waveforms by bicubic spline interpolation to five times the sampling rate $f_{\mathrm{rate}}$, aligned on their local minima, and then resampled at $f_{\mathrm{rate}}$. We then performed a Principal Component Analysis (PCA) on these centered and aligned waveforms and kept only the first $N_{\mathrm{PCA}}$ principal components. In all the calculations we used default values of $N_\mathrm{p} = 10000$ and $N_{\mathrm{PCA}} = 5$. These principal components were used during the clustering step.

## Clustering
The goal of the clustering step is to construct a dictionary of templates. As opposed to former clustering approaches of spike sorting (*Quiroga et al., 2004*; *Harris et al., 2000*; *Kadir et al., 2014*), because this clustering step is followed by a template matching, we do not need to perform the clustering on all the spikes.

**Table 1.** Table of all the variables and notations found in the algorithm.

| Variable | Explanation | Default value |
|---|---|---|
| | Generic notations | |
| $N_{\text{elec}}$ | Number of electrodes | |
| $\mathbf{p_k}$ | Physical position of electrode $k$ [$\mu$m] | |
| $G_k$ | Ensemble of nearby electrodes for electrode $k$ [$\mu$m] | |
| $N^k_{\text{neigh}}$ | Cardinal of $G_k$ | |
| $\theta_k$ | Spike detection threshold for electrode $k$ [$\mu$V] | |
| $\mathbf{s(t)}$ | Raw data [$\mu$V] | |
| $\mathbf{w}_j(t)$ | First component of the template for neuron $j$ [$\mu$V] | |
| $\mathbf{v}_j(t)$ | Second component of the template for neuron $j$ [$\mu$V] | |
| $f_{\text{rate}}$ | Sampling frequency of the signal [Hz] | |
| | Preprocessing of the data | |
| $f_{\text{cut}}$ | Cutoff frequency for butterworth filtering | 100 Hz |
| $N_t$ | Temporal width for the templates | 5 ms |
| $r_{\text{max}}$ | Spatial radius for the templates | 250 $\mu$m |
| $\lambda$ | Gain for threshold detection for channel $k$ ($\theta_k$) | 6 |
| $N_p$ | Number of waveforms collected per electrode | 10000 |
| $N_{\text{PCA}}$ | Number PCA features kept to describe a waveform | 5 |
| | Clustering and template estimation | |
| $\mathbf{x}^k_{1,..l}$ | $l$ spikes peaking on electrode $k$ and projected after PCA | |
| $\rho^k_l$ | Density around $\mathbf{x}^k_l$ | |
| $\delta^k_l$ | Minimal distance from $\mathbf{x}^k_l$ to spikes with higher densities | |
| $N_{\text{spikes}}$ | Number of spikes collected per electrode for clustering | 10000 |
| $N_{\text{PCA}_2}$ | Number of PCA features kept to describe a spike | 5 |
| $S$ | Number of neighbors for density estimation | 100 |
| $N^{\text{clusters}}_{\text{max}}$ | Maximal number of clusters per electrode | 10 |
| $\zeta$ | Normalized distance between pairs of clusters | |
| $\sigma_{\text{similar}}$ | Threshold for merging clusters on the same electrode | 3 |
| $\alpha_m$ | Centroid of the cluster $m$ | |
| $\gamma_m$ | Dispersion around the centroid $\alpha_m$ | |
| $\eta$ | Minimal size of a cluster (in percent of $N_{\text{spikes}}$) | 0.005 |
| $[a_{\text{min}}, a_{\text{max}}]$ | Amplitudes allowed during fitting for a given template | |
| | Dictionary cleaning | |
| $CC_{\text{max}}(m, n)$ | Max over time for the Cross-correlation between $\mathbf{w}_m$ and $\mathbf{w}_n$ | |
| $cc_{\text{similar}}$ | Threshold above which templates are considered as similar | 0.975 |
| | Template matching | |
| $a_{ij}$ | Product between $\mathbf{s(t)}$ and $\mathbf{w_j}$ (normalized) at time $t_i$ | |
| $b_{ij}$ | Same as $a_{ij}$ but for the second component $\mathbf{v_j}$ | |
| $n_{\text{failures}}$ | Number of fitting attempts for a given spike time | 3 |
| | Automated merging | |
| $cc_{\text{merge}}$ | Similarity threshold to consider neurons as a putative pair | 0.8 |
| $r_{m,n}(t)$ | Cross correlogram between spikes of unit $m$ and $n$ | |
| $\phi(m, n)$ | Geometrical mean of the firing rates for units $m$ and $n$ [Hz$^2$] | |
| $\phi_{\text{merge}}$ | Maximal value for the dip in the cross correlogram at time 0 | 0.1 [Hz$^2$] |

DOI: https://doi.org/10.7554/eLife.34518.007

## Masking

We first randomly collected a subset of many spikes $t_i$ to perform the clustering. To minimize redundancy between collected spikes, we prevented the algorithm to have two spikes peaking on the same electrode separated by less than $N_t/2$.

## Pre-clustering of the spikes

Trying to cluster all the spikes from all the electrodes at once is very challenging, because they are numerous and live in a high dimensional space. We used a divide and conquer approach to parallelize this problem (*Marre et al., 2012*; *Swindale and Spacek, 2014*). Each time a spike was detected at time $t_i$, we searched for the electrode $k$ where the voltage $\mathrm{s}(t_i)$ has the lowest value, that is such that. For every electrode $k$ we collected a maximum of $N_{\mathrm{spikes}}$ spikes (set to 10,000 by default) peaking on this electrode. Each of these spikes is represented by a spatio-temporal waveform of size $N_t \times N_{\mathrm{neigh}}^k$, where $N_{\mathrm{neigh}}^k$ is the number of electrodes in the vicinity of electrode $k$, that is the number of elements in $G^k$. Note that, in the following we did not assume that spikes are only detected on a single electrode. We used the information available on all the neighboring electrodes.

We projected each temporal waveform on the PCA basis, estimated earlier, to reduce the dimensionality to $N_{\mathrm{PCA}} \times N_{\mathrm{neigh}}^k$. During this projection, the same up-sampling technique described in the Pre-processing was used. Each spike was then represented in a space with $N_{\mathrm{PCA}} \times N_{\mathrm{neigh}}^i$ dimensions. To reduce dimensionality even further before the clustering stage, for every electrode $k$ we performed a PCA on the collected spikes and kept only the first $N_{\mathrm{PCA}_2}$ principal components (in all the paper, $N_{\mathrm{PCA}_2} = 5$). Therefore, we performed a clustering in parallel for every electrode on at max $N_{\mathrm{spikes}}$ described in a space of $N_{\mathrm{PCA}_2}$-dimension.

## Clustering by search of local density peaks

To perform the clustering, we used a modified version of the algorithm published in (*Rodriguez and Laio, 2014*). We note the spikes $\{1,..,l\}$ associated with electrode $k$ (and projected on the second PCA basis) as vectors $\mathbf{x}_{\{1,..,l\}}^k$ in a $N_{\mathrm{PCA}_2}$ dimensional space. For each of these vectors, we estimated $\rho_l^k$ as the mean distance to the $S$ nearest neighbors of $\mathbf{x}_l^k$. Note that $1/\rho_l^k$ can be considered as a proxy for the density of points. $S$ is chosen such that $S = \epsilon N_{\mathrm{spikes}}$, with $\epsilon = 0.01$. In our settings, since $N_{\mathrm{spikes}} = 10000$ then $S = 100$. This density measure turned out to be more robust than the one given in the original paper and rather insensitive to changes in $\epsilon$. To avoid a potentially inaccurate estimation of the $\rho_l^k$ values, we collected iteratively additional spikes to refine this estimate. Keeping in memory the spikes $\mathbf{x}_l^k$, we searched again in the data $N_{\mathrm{spikes}}^k$ different spikes andused them only to refine the estimation of $\rho_l^k$ of our selected points $\mathbf{x}_l^k$. This refinement gave more robust results for the clustering and we performed 3 rounds of this new search. Then, for every point $\mathbf{x}_l^k$, we computed $\delta_l^k$ as the minimal distance to any other point $\mathbf{x}_{m,m\neq l}^k$ such that $\rho_m^k \leq \rho_l^k$. This corresponds to the distance to the nearest point with a higher density. The intuition of the algorithm is that the centroids should be points with a high density (i.e. low $\rho$) and far apart from each others (high $\delta$).

## Centroids and cluster definition

To define the centroids we ranked the points as a function of the ratios $\delta/\rho$ and we detected the best $N_{\mathrm{clusters}}^{\max}$ points as putative centroids. By default $N_a^{\max} thrmclusters = 10$. Intuitively, this parameter corresponds to the maximal number of cells that will peak on any given electrode. It can be seen as an upper bound of the ratio between the number of cells and the number of electrodes. ructure recorded, the density of cells and the density of the electrodes, this number can be varied. Clusters were formed by assigning each point to one of the selected centroids following an iterative rule, going from the points of lowest $\rho$ to the points of highest $\rho$: each point was assigned to the same cluster as the closest point with a lower $\rho$ (*Rodriguez and Laio, 2014*). Thanks to this ordering, we started by assigning the points of highest density to the nearest centroid, and then assigned the next points to the nearest point with higher density, which has been already assigned to a cluster. We created $N_{\mathrm{clusters}}^{\max}$ clusters. Once this is done, we iteratively merged pairs of clusters that were too similar to each others.

## Merging similar clusters

We computed a normalized distance $\zeta$ between each pair of clusters. The center $\alpha_m$ of each cluster was defined as the median of all the points composing this cluster. For each pair of cluster $(m, n)$, we then projected all the points foreach of them onto the axis joining the two centroids $\alpha_m - \alpha_n$. We defined the dispersions around the centroids $\alpha_m$ as $\gamma_m = \mathrm{MAD}(\mathbf{x}_m \cdot (\alpha_m - \alpha_n))$, where $\cdot$ is the scalar product of two vectors. The normalized distance is:

$$\zeta(m,n) = \frac{\|\alpha_m - \alpha_n\|}{\sqrt{\gamma_m^2 + \gamma_n^2}} \tag{2}$$

We then iteratively merged all clusters $(m, n)$ such that $\zeta(m,n) \leq \sigma_{\mathrm{similar}}$. At the end of the clustering, every cluster with less than $\eta N_{\mathrm{spikes}}^i$ was discarded. In all the manuscript we used $\sigma_{\mathrm{similar}} = 3$, $N_{\mathrm{clusters}}^{\max} = 10$, and $\eta = 0.005$. We chose $N_{\mathrm{clusters}}^{\max} = 10$ because we never see more than 10 neurons peaking on the same electrode, and this approximately corresponds to the ratio between density of observable cells and density of electrodes.

## Template estimation

At the end of the clustering phase, pooling the clusters obtained from every electrode, we obtained a list of clusters. Each cluster $m$ is defined by a list of spike times $t_{\{1,\dots,l\}}^m$. During this phase we limited the number of spike times per template to a maximal value of 500 to avoid memory saturation, because we had to keep in memory these 500 snippets.

We computed the first component from the raw data as the point-wise median of all the waveforms belonging to the cluster: $\mathbf{w}_m(t) = \mathrm{med}_l \mathbf{s}(t_l^m + t)$. Note that $\mathbf{w}_m$ is only different from zero on the electrodes close to its peak (see *Figure 1C*). This information is used internally by the algorithm to save templates as sparse structures. We set to 0 all the electrodes $k$ where $\|\mathbf{w}_m^k(t)\| < \theta_k$, where $\theta_k$ is the detection threshold on electrode $k$. This allowed us to remove electrodes without discriminant information and to increase the sparsity of the templates.

We then computed the projection of all snippets in the space orthogonal to the first component: $\forall l, \mathbf{q}_l = \mathbf{s}(t_l^m) - \beta_l \mathbf{w}_m$, with $\beta_l = \frac{\mathbf{s}(t_l^m) \cdot \mathbf{w}_m}{\|\mathbf{w}_m\|}$. The $\mathbf{q}$ are the projections of the waveforms in a space orthogonal to $\mathbf{w}_m$. We estimated the second component of the template $\mathbf{v}_m(t)$ as the direction of largest variance in this orthogonal space (i.e. the first principal component of $\mathbf{q}_l$).

From the first components $\mathbf{w}_m$, we also computed its minimal and maximal amplitudes $a_m^{\min/\max}$. If $\mathbf{w}_m$ is the normalized template, such that $\mathbf{w}_m = \mathbf{w}_m / \|\mathbf{w}_m\|$, we computed

$$a_h^{\min} = \mathrm{med}_l \mathbf{s}(t_l^m) . \hat{\mathbf{w}}_m - 5\mathrm{MAD}_l(\mathbf{s}(t_l^m) . \hat{\mathbf{w}}_m) \tag{3}$$

$$a_h^{\max} = \mathrm{med}_l \mathbf{s}(t_l^m) . \hat{\mathbf{w}}_m + 5\mathrm{MAD}_l(\mathbf{s}(t_l^m) . \hat{\mathbf{w}}_m)$$

Those boundaries are used during the template matching step (see below). The factor five allows most of the points to have their amplitude between the two limits.

## Removing redundant templates

To remove redundant templates that may be present in the dictionary because of the divide and conquer approach (for example a neuron between two electrodes would give rise to two very similar templates on two electrodes), we computed for all pairs of templates in the dictionary $CC_{\max}(m,n) = \max_t CC(\mathbf{w}_m, \mathbf{w}_n)$, where $CC$ stands for normalized cross-correlation. If $CC_{\max}(m,n) \geq cc_{\mathrm{similar}}$, we considered these templates to be equivalent and they were merged. In all the following, we used $cc_{\mathrm{similar}} = 0.975$. Note that we computed the cross-correlations between normalized templates such that two templates that have the same shape but different amplitudes are merged. Similarly, we searched if any template $\mathbf{w}_p$ could be explained as a linear combination of two templates in the dictionary. If we could find $\mathbf{w}_m$ and $\mathbf{w}_n$ such that $CC(\mathbf{w}_p, \mathbf{w}_m + \mathbf{w}_n) \geq cc_{\mathrm{similar}}$, $\mathbf{w}_p$ was considered to be a mixture of two cells and was removed from the dictionary.

## Template matching

At the end of this 'template-finding' phase we have found a dictionary of templates $(\mathbf{w}, \mathbf{v})$. We now need to reconstruct the signal s by finding the amplitudes coefficients $a_{ij}$ and $b_{ij}$ described in *Equation 1*. Because at a given spike time $t_i$ it is likely that only a couple of cells will fire a spike, most $a_{ij}$ and $b_{ij}$ in this equation are equal to 0. For the other ones most $a_{ij}$ values are around one because a spike usually appears on electrodes with an amplitude close to the average first component $\mathbf{w}$. In this template matching step, all the other parameters have been determined by template extraction and spike detection, so the purpose is only to find the values of these amplitudes. Note that the spike times $t_i$ were detected using the method described above and include all the threshold crossing voltages that are local minima. Each true spike can be detected over several electrodes at slightly different times such that there are many more $t_i$ than actual spikes. With this approach, we found that there was no need to shift templates before matching them to the raw data.

To match the templates to the data we used an iterative greedy approach to estimate the $a_{ij}$ for each $t_i$, which bears some similarity to the matching pursuit algorithm (*Mallat and Zhifeng Zhang, 1993*). The fitting was performed in blocks of putative spike times,$\{t_i\}$, that were successively loaded in memory. The size of one block was typically one second, which includes a lot of spike times, and is much larger than a single snippet. The snippets were thus not fitted independently from each other. The successive blocks were always overlapping by two times the size of a snippet and we discarded the results obtained on the borders to avoid any error of the template matching that would be due to a spike split between two blocks. Such an approach allowed us to easily split the workload linearly among several processors.

Each block of raw data s was loaded and template-matching was performed according to the following steps:

1. Estimate the normalized scalar products $\mathbf{s}(t) \cdot \mathbf{w}_j(t - t_i)$ for each template $j$ and putative spike time $t_i$, for all the $i$ and $j$ in the block of raw data.
2. Choose the $(i, j)$ pair with the highest scalar product, excluding the pairs $(i, j)$ which have already been tried and the $t_i$'s already explored (see below).
3. Set $a_{ij}$ equal to the amplitude value that best fits the raw data: $a_{ij} = \frac{\mathbf{s}(t).\mathbf{w}_j(t-t_i)}{\|\mathbf{w}_j(t-t_i)\|}$.
4. Check if the $a_{ij}$ amplitude value is between $a_j^{\min}$ and $a_j^{\max}$.
5. If yes, accept this value, subtract the scaled template from the raw data: $\mathbf{s}(t) \leftarrow \mathbf{s}(t) - a_{ij}\mathbf{w}_j(t - t_i)$. Then set $b_{ij}$ equal to the amplitude value that best fits the raw data with $\mathbf{v}_j$, and subtractit too. Then return to step one to re-estimate the scalar products on the residual.
6. Otherwise increase by one $n_i$, which counts the number of times any template has been rejected for spike time $t_i$.
   1. If $n_i$ reaches $n_{\text{failures}} = 3$, label this $t_i$ as 'explored'. If all $t_i$ have been explored, quit the loop.
   2. Otherwise return to step one and iterate.

The parameters of the algorithm were the amplitude thresholds $a_j^{\min}$ and $a_j^{\max}$, computed as described in the section Template Estimation.

## Automated merging

For the template similarity, we computed, for every pair of templates $m$ and $n$, $CC_{\max}(m, n) = \max_t CC(\mathbf{w}_m, \mathbf{w}_n)$ (where $CC$ is the normalized cross-correlation between the two templates - see above forthe definition). To quantify the dip in the cross-correlogram, we binned the spike trains obtained for templates $m$ and $n$ with 2 ms bin size, and estimated the cross correlogram $r_{m,n}(\tau)$ between unit $m$ and unit $n$, defined as $\langle \sigma_m(t)\sigma_n(t + \tau) \rangle_t$. $\sigma_m(t)$ is the number of spikes of unit $m$ in time bin $t$. We then estimated the dip as the difference between the value of the cross-correlogram at time 0 ms and the geometrical mean of the firing rates, that is $\phi(m, n) = \langle \sigma_m(t) \rangle_t \langle \sigma_n(t) \rangle_t$. This geometrical mean would be the value of the cross-correlogram if the two spike trains were independent. The dip is therefore estimated as

$$\langle \sigma_m(t) \rangle_t \langle \sigma_n(t) \rangle_t - \langle \sigma_m(t)\sigma_n(t + \tau) \rangle_t \tag{4}$$

We plotted the values of the estimated dip, the template similarity and the geometrical mean of

the firing rates for each pair in a Graphical User Interface (GUI). The user can quickly define at once a whole set of pairs that need to be merged. After merging a subset of the pairs, quantities $CC_{\max}$ and $\phi$ are re-computed, until the user decides to stop merging (see *Figure 4*).

If the two spike trains from templates $m$ and $n$ correspond to the same cell, there should be no refractory spike trains. The cross-correlogram value should be close to 0 and the dip estimation should therefore be close to the geometrical mean of the firing rates. To formalize this intuition and fully automate the merging, we decided to merge all the pairs $(m, n)$ such that:

$$CC_{\max}(m,n) > cc_{\mathrm{merge}} \, \mathrm{and} \, \langle \sigma_m(t)\sigma_n(t+\tau)\rangle_t \le \phi_{\mathrm{merge}} \tag{5}$$

with $cc_{\mathrm{merge}} = 0.8$ and $\phi_{\mathrm{merge}} = 0.1$. This corresponds to merging all the highly similar pairs above a line parallel to the diagonal (see *Figure 4A,B*, gray area). With these two parameters we could automate the merging process.

## Simulated ground truth tests

In order to assess the performance of the algorithm, we injected new templates in real datasets (see *Figure 3D*). To do so, we ran the algorithm on a given dataset and obtain a list of putative templates $\mathbf{w}_{j \in \{1,\ldots N\}}$. Then, we randomly selected some of those templates $\mathbf{w}_j$ and shuffled the list of their electrodes before injecting them elsewhere in the datasets at controlled firing rates (*Harris et al., 2000*; *Rossant et al., 2016*; *Kadir et al., 2014*; *Segev et al., 2004*; *Marre et al., 2012*; *Chung et al., 2017*). This enabled us to properly quantify the performance of the algorithm. In order not to bias the clustering, when a template $\mathbf{w}_j$ was selected and shuffled as a new template $\bar{\mathbf{w}}_k$ centered on a newelectrode $k$, we ensured that the injected template was not too similar to one that would already be in the data: $\forall h \in \{1, \ldots N\}, \max_t CC(\mathbf{w}_h, \bar{\mathbf{w}}_k) \le 0.8$. Before being injected, $\bar{\mathbf{w}}_k$ was normalized such that $\min_t \bar{\mathbf{w}}_k = \alpha_k \theta_k$. $\alpha_k$ is the relative amplitude, expressed as function of $\theta_k$, the detection threshold on the electrode where the template is peaking. If $\alpha_k \le 1$ the template is smaller than spike threshold, and its spikes should not be detected; if $\alpha_k \ge 1$ the spikes should be detected. In *Figure 3G*, we injected the artificial templates into the data such that they were all firing at 10 Hz, but with a controlled correlation coefficient $c$ that could be varied (using a Multiple Interaction Process [*Kuhn et al., 2003*]). This parameter $c$ allowed us to quantify the percentage of pairwise correlations recovered by the algorithm for overlapping spatio-temporal templates.

## Performance estimation

### Estimation of false positives and false negatives

To quantify the performance of the algorithm we matched the spikes recovered by the algorithm to the real ground-truth spikes (either synthetic or obtained with juxta-cellular recordings). A spike was considered to be a match if it had a corresponding spike in the ground truth at less than 2 ms. Spikes in the ground-truth datasets that had no matches in the spike sorting results in a 2 ms window were labeled as 'false negatives', while those that are not present while the algorithm detected a spike were 'false positives'. The false-negative rate was defined as the number of false negatives divided by the number of spikes in the ground truth recording. The false-positive rate was defined as the number of false positives divided by the number of spikes in the spike train extracted by the algorithm. In the paper, the error is defined as mean of the false negative and the false positive rates (see *Figures 2* and *3*). Note that to take into account the fact that a ground-truth neuron could be split into several templates at the end of the algorithm, we always compared the ground-truth cells with the combination of templates that minimized the error.

### Theoretical estimate

To quantify the performance of the software with real ground-truth recordings (see *Figure 2*) we computed the Best Ellipsoidal Error Rate (BEER), as described in (*Harris et al., 2000*). This BEER estimate gave an upper bound on the performance of any clustering-based spike sorting method using elliptical cluster boundaries. After thresholding and feature extraction, snippets were labeled according to whether or not they contained a true spike. Half of this labeled data set was then used to train a perceptron whose decision rule is a linear combination of all pairwise products of the features of each snippet. If $\mathbf{x}_i$ is the $i$-th snippet, projected in the feature space, then the optimized function $f(\mathbf{x})$ is:

$$f(\mathbf{x}) = \mathbf{x}^T A \mathbf{x} + b^T \mathbf{x} + c \qquad (6)$$

We trained this function $f$ by varying $A$, $b$ and $c$ with the objective that $f(\mathbf{x})$ should be +1 for the ground truth spikes, and $-1$ otherwise. These parameters were optimized by a stochastic gradient descent with a regularization constraint. The resulting classifier was then used to predict the occurrence of spikes in the snippets in the remaining half of the labeled data. Only the snippets where $f(\mathbf{x}) > 0$ were predicted as true spikes. This prediction provided an estimate of the false-negative and false-positive rates for the BEER estimate. The mean between the two was considered to be the BEER error rate, or 'Optimal Classifier Error'.

## Decimation of the electrodes

In order to increase the number of data points for the comparison between our sorting algorithm and the nonlinear classifiers defined by the BEER metric (see *Figure 2*), we ran the analysis several times on the same neurons, but removing some electrodes, to create recordings at a lower electrode density. We divided by a factor 2 or 4 the number of electrodes in the 252 in vitro Multielectrode Array or the 128 in vivo silicon probe.

## Hardware specifications

The comparison between Kilosort (*Pachitariu et al., 2016*) and SpyKING CIRCUS was performed on a desktop machine with 32 Gb RAM and eight cores (proc Intel Xeon(R) CPU E5-1630 v3 @ 3.70 GHz). The GPU used was a NVIDIA Quadro K4200 with 4 Gb of dedicated memory.

## Implementation and source code

SpyKING CIRCUS is a pure Python package, based on the python wrapper for the Message Passing Interface (MPI) library (*Dalcin et al., 2011*) to allow parallelization over distributed computers, and is available with its full documentation at http://spyking-circus.rtfd.org. Results can easily be exported to the kwik or phy format (*Rossant and Harris, 2013*). All the datasets used in this manuscript are available on-line, for testing and comparison with other algorithms (*Spampinato et al., 2018* ).

## Acknowledgements

We would like to thank Charlotte Deleuze for her help with the in vitro juxtacellular recordings, and Steve Baccus and Sami El Boustani for insightful discussions. We also would like to thank Kenneth Harris, Cyrille Rossant and Nick Steimetz for feedbacks and the help with the interface to the phy software. This work was supported by ANR-14-CE13-0003 to PY, ANR TRAJECTORY, ANR OPTIMA, the French State program Investissements d'Avenir managed by the Agence Nationale de la Recherche [LIFESENSES: ANR-10-LABX-65], an EC grant from the Human Brain Project (FP7-720270), and NIH grant U01NS090501 to OM, ERC Starting Grant (309776) to JD and Foundation Figthing Blindness to SP.

## Additional information

### Funding

| Funder | Grant reference number | Author |
| --- | --- | --- |
| Agence Nationale de la Recherche | TRAJECTORY | Olivier Marre |
| European Commission | ERC StG 309776 | Jens Duebel |
| National Institutes of Health | U01NS090501 | Olivier Marre |
| Foundation Fighting Blindness | | Serge Picaud |
| Agence Nationale de la Recherche | ANR-14-CE13-0003 | Pierre Yger |
| Agence Nationale de la Recherche | ANR-10-LABX-65 | Serge Picaud |

| | | |
|---|---|---|
| European Commission | FP7-720270 | Olivier Marre |

The funders had no role in study design, data collection and interpretation, or the decision to submit the work for publication.

## Author contributions

Pierre Yger, Conceptualization, Data curation, Software, Formal analysis, Supervision, Validation, Investigation, Visualization, Methodology, Writing—original draft, Project administration, Writing—review and editing; Giulia LB Spampinato, Conceptualization, Resources, Data curation, Formal analysis, Validation, Investigation, Visualization, Methodology, Writing—original draft, Project administration, Writing—review and editing; Elric Esposito, Conceptualization, Resources, Data curation, Validation, Investigation, Visualization, Methodology, Project administration, Writing—review and editing; Baptiste Lefebvre, Conceptualization, Software, Formal analysis, Validation, Investigation, Visualization, Methodology, Project administration, Writing—review and editing; Stéphane Deny, Resources, Data curation; Christophe Gardella, Data curation, Software, Formal analysis, Validation, Writing—review and editing; Marcel Stimberg, Software, Project administration; Florian Jetter, Resources; Guenther Zeck, Resources, Writing—review and editing; Serge Picaud, Conceptualization, Supervision, Funding acquisition, Project administration, Writing—review and editing; Jens Duebel, Supervision, Funding acquisition, Project administration, Writing—review and editing; Olivier Marre, Conceptualization, Resources, Data curation, Software, Formal analysis, Supervision, Funding acquisition, Validation, Investigation, Visualization, Methodology, Writing—original draft, Project administration, Writing—review and editing

## Author ORCIDs

Pierre Yger [iD] http://orcid.org/0000-0003-1376-5240
Christophe Gardella [iD] https://orcid.org/0000-0003-3204-9012
Marcel Stimberg [iD] http://orcid.org/0000-0002-2648-4790
Olivier Marre [iD] http://orcid.org/0000-0002-0090-6190

## Ethics

Animal experimentation: Experiments were performed in accordance with institutional animal care standards, using protocol (#00847.02) of the Institut de la Vision (Agreement number A751202). The protocol was approved by the Charles Darwin ethic committee (CEEACD/N°5)

## Decision letter and Author response

Decision letter https://doi.org/10.7554/eLife.34518.014
Author response https://doi.org/10.7554/eLife.34518.015

# Additional files

## Supplementary files

• Transparent reporting form
DOI: https://doi.org/10.7554/eLife.34518.008

## Major datasets

The following dataset was generated:

| Author(s) | Year | Dataset title | Dataset URL | Database, license, and accessibility information |
|---|---|---|---|---|
| Giulia LB Spampinato, Elric Esposito, Pierre Yger, Jens Duebel, Serge Picaud, Olivier Marre | 2018 | Ground truth recordings for validation of spike sorting algorithms | http://dx.doi.org/10.5281/zenodo.1205233 | Publicly available at Zenodo |

The following previously published dataset was used:

| Author(s) | Year | Dataset title | Dataset URL | Database, license, and accessibility information |
|---|---|---|---|---|
| Neto JP, Lopes G, Frazao J, Nogueira J, Lacerda P, Baiao P, Aarts A, Andrei A, Musa s, Fortunato E, Barquinha P, Kampff AR | 2016 | Ground-Truth data from silicon polytrodes | http://www.kampff-lab.org/validating-electrodes/ | Directly available for download |

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
