## [Decision Letter]

[Editors’ note: a previous version of this study was rejected after peer review, but the authors submitted for reconsideration. The first decision letter after peer review is shown below.]

Thank you for submitting your work entitled "A spike sorting toolbox for up to thousands of electrodes validated with ground truth recordings in vitro and in vivo" for consideration by *eLife*. Your article has been reviewed by three peer reviewers, and the evaluation has been overseen by a Reviewing Editor and a Senior Editor.

Your work is of particular interest based on the rarity of ground truth data for spike sorting. Unfortunately, the reviewers had mixed feeling about the depth and clarity of the data and voice some significant concerns that speak against publication. The chief concern is that the ground truth data for the MEA is limited to the 7 cells for sorting errors less than 20% (Figure 2C; see comment by reviewer 1). In fact, the example (presumably one of the seven points) in Figure 2B suggests that a given cell goes not contribute a waveform to many electrodes (as implied in Figure 1; see additional comment by reviewer 1). Thus the necessity as well as utility of the algorithm seems unmotivated as it stands. Another weakness, discussed by reviewers 1 and 2, is that the merging of clusters is "by hand", although there is precedent in the literature (Fee et al. 1997) for automatic merging based on avoiding events at equal time in the autocorrelation of the merged clusters. Finally, all reviewers and the Reviewing Editor find the presentation confusing or lacking. Material is not presented in a logical order, details of the algorithm as glossed over, etc.

All reviewers felt that the topic of the manuscript is timely. We understand that you may wish to take this manuscript elsewhere at this time. On the other hand, if you agree with the reviewers, we would be willing to consider a revised manuscript with a larger useful data set and with a means for automation of the merging of clusters, without assurance that the revision would be successful.

*Reviewer #1:*

These authors report their method to do classic spike sorting, creation of single neuron spike time records, from multisite recording. They report results on both multielectrode arrays, used to record in vitro from retinal tissue, and Si probes, used to record in vivo, most frequently from rodents. Their method has been previewed on bioRxiv for some time so there is some public information comparing performance of this method to existing and other recently developed packages. While the technical treatment is thorough, I find the paper confusing, in large part because of the bifurcation of the Materials and methods and the remainder of the paper. To begin, the authors say "those snippets are detected, they are projected in a feature space of lower dimension". What feature space. Ten pages later I find out PCA. How many dimensions, either 5 or 6 depending on the parts of the algorithm. Without comment, the templates are 57 sites. This seems exceptionally large, perhaps usual for MEA recording. The choice of size deserves a clear comment.

I commend the authors on their contribution of "ground truth" MEA data. The field is seriously deficient in reference data and this deficiency impairs progress in the computational problem at hand. Figure 2A and B are commendably clear. I find 2C to be puzzling. I would assume only the inset of retina in vitro is of any interest to the practitioners of neural recording. Error rates above 20% are of no practical utility but occupy 95% of the plot. Alternatively only 1 data point in the in vivo plat is useful. This result is not surprising given the traces in 2B. Peak to peak noise is ~10-12 uV. The usual peak threshold is 3.5-5 x RMS or 12-15 uV. As such, one would not expect to sort signal below that amplitude. With the authors template matching routine, it can be argued that smaller signals can be found but they are buried in noise, again as Figure 2C seems to illustrate. I am left wondering what 2C teaches, except, that sorting has failed in most of the conditions plotted. Does this not call into question the success of the strategy of finding template pools at high amplitude then matching those templates to the raw traces? If that is not the lesson, then the authors need to clarify the value of plotting so many failed sorting exercises. I fail to understand the value of a 50% classification error that matches the optimal classification rate. I would conclude the experiment has failed to produce adequate usable data. Further, the authors choose to compare their sorting accuracy to a previously defined metric, Best Ellipsoidal Error Rate. Harris, 2000, the original source defines this measure as "This measure was designed to estimate the optimal clustering performance for a given set of feature vectors by searching over all possible ellipsoidal cluster boundaries." Is not the principal weakness of the methods of Harris reported there and later an assumption that a cluster is well described by ellipsoidal boundaries? The authors should justify this assumption in the reference data sets. I see no a priori reason that their data should be so constrained, although the "ground spikes" may be.

In the Discussion, especially Figure 3, trends are difficult to understand. For 3A, the plot seems to argue a superlinear dependence on processor count, 6 processors taking 1/30th the time of 2 processors. This seems implausible. Would this plot be more informative on a log time scale without a trend-line. The current plot implies that 10 processors would digest the data in negative time! 3B is much more informative, showing 400 channels would take ~20 hours of computation for 1 hour of recording. While an improvement over many current methods, is not the method of Pachatariu et al., reference in this manuscript as a bioRxiv but now also published in Advances In Neural Information Processing Systems 4448-4456 (2016). The GPU based system reported their quotes "First, we timed the algorithm on several large scale datasets. The average run times for 32, 128 and 384 channel recordings were 10, 29 and 140 minutes respectively, on a single GPU-equipped workstation". While they did not quote the duration of the recording or the total number of neuronal event (a better metric of the computation burden) it seems far faster than the ~30 hours in the present manuscript. Both of the packages are available in the public domain so a direct comparison is a reasonable request. In Figure 3E, the authors plot data ranging from 0 to 15% (including error bars) to 50% full scale. Why not use the full plot height for 15%. Same comment for 3F.

Finally, the authors freely admit their strategy over segments but offer a combination of an automated merging tool as well as a "dedicated GUI" for manual merging. This reviewer did not implement that interface so I cannot comment directly on performance or ease of use. In the Materials and methods section, they describe in detail the method for correlation, but they do no show the error rate as a function of the parameters used for that merging.

In summary, I am undecided. The authors use an admirable amount to "ground truth" data, both their own and that available publicly. Unfortunately, all of these data are inadequate to the problem since the known cells are so few and in the experimental data of such low amplitude yield a residual question of performance remains. This manuscript reports a novel combination of strategies but does not appear to exceed in performance or speed other recently published packages. More than one option is probably justified based on different needs of data sets, but that optimum path is difficult to discern from this manuscript of other published data. Reporting the method is a service to the community and justified in journal space, especially for an online journal such as *eLife*.

*Reviewer #2 (General assessment and major comments (Required)):*

In this report, the authors present a new approach to spike sorting that they claim is scalable to thousands of electrodes. This a very timely report as there is renewed interest in spike sorting thanks to the new electrode technologies. However, I find this report to be mixed in terms of strength of the proposed solution and clarity with which the underlying claims are made.

The strength of the proposed solution is significantly under-cut by the need for manual cluster merging. This is a really major limitation. Clustering of thousands of electrodes has to be automatic if it is to be genuinely scalable. I can imagine it would take a long time for someone to check thousands of recordings and merge, especially if they are not an expert.

The authors cite several algorithms that are designed to scale and says they are untested. This is a fair point, but why don't they test them with their ground truth data and compare the results? This would seem to be an important issue for the reader to assess the value of the proposed work in the context of the cited literature.

The Introduction implies that the spike sorting problem has been addressed for small numbers of contacts, but does not include any citations to support the point. What forms of the classical approach that the authors are referring to? I would claim that despite a great deal of attention it is not the case that low-density sorting has been solved. The high density electrode case that the authors focus on here presents new challenges but the authors need to acknowledge the known limitations of low-density approaches.

A novelty of the proposed approach is that it claims to address the superposition problem but how well this works is not demonstrated or discussed. I think the work would be strengthened if this aspect received more attention.

*Reviewer #3:*

This manuscript presents a novel resource for automated spike sorting on data from large-scale multi-electrode recordings. The authors present new strategies for tackling large recordings with hundreds to thousands of electrodes while being able to integrate information from multiple electrodes and to disentangle overlapping spikes from multiple neurons. Thus, the resource addresses fundamental issues of spike sorting. Given the current development of recording systems with hundreds to thousands of electrodes and the paucity of methods targeted to such data sets, the presented method should therefore make a timely and very strong contribution to the field of multi-electrode recordings.

The approach appears to be well thought-through and apparently guided by much practical experience with spike-sorting issues. The manuscript is accompanied by freely available software that can be downloaded from the authors' website, and the software appears to be well documented and versatile in that it can be run on different operating systems, using different hardware configurations, with straightforward parallelization. Moreover, the authors test their spike-sorting approach on ground-truth data by combining their extracellular multi-electrode recordings with the intracellular recording of individual neurons, a laudable effort. They generally find that their automated spike sorting performs nearly as well as an "optimal classifier", obtained by supervised learning of an elliptical decision boundary in the space of raw electrode signals. The authors also make this ground-truth data available, which will be a great asset available to future developments of spike sorting approaches.

My only concerns therefore relate to the presentation of the material. Given that this is a presentation of a spike sorting resource, it seems mandatory that the computational approach and the technical aspects of the algorithm be explained as thoroughly and clearly as possible so that readers and users can fully understand the approach and adjust it to their own needs by tuning the appropriate parameters. In the Results part, the explanation of the algorithm is rather brief, and it would here be useful to provide a bit more of an intuitive description of the approach (see detailed points below). The Materials and methods section, on the other hand, appears to contain some inaccuracies and small omissions of detail that make it hard to follow some of the steps.

1) The Results part is a bit sparse on explaining the approach, although this would be a nice opportunity to lay out concepts without the need for technical details. For example, it might be useful to provide some brief explanations of why limiting the clustering to the search for centroids is less demanding and what the approach in density-based clustering is. Furthermore, some explanation of what is meant by "clustering each group in parallel" would be helpful, including the strategy for finding the right number of clusters and merging templates and units when necessary.

2) One aspect that doesn't become clear in the main part is to what extent merging of units generally needs to be performed and how it was done for the results of Figures 2 and 3. The Materials and methods section states that several templates were used for comparison with ground-truth data. It would be useful to obtain some more information about this. Was this merging done with the GUI described in the Discussion? Was the merging done "blindly", that is, before comparison with the ground-truth data, or retrospectively? Was merging the norm or the exception?

3) A potential issue with automated spike sorting is that it might be difficult to judge the sorting quality of the obtained units, that is, how reliable the units are and to what degree one may expect contamination from other cells or missed spikes. Does the resource provide for any quality control? Useful simple statistics might be the normalized spike amplitude, the distance from other templates, the quality of the fits in the template matching phase, and the variability in the fitting parameters, but maybe the authors could also point out other ways of assessing sorting quality from their own experience of using their software.

[Editors’ note: what now follows is the decision letter after the authors submitted for further consideration.]

Thank you for resubmitting your article "A spike sorting toolbox for up to thousands of electrodes validated with ground truth recordings in vitro and in vivo" for consideration by *eLife*. Your article has been reviewed by three peer reviewers, and the evaluation has been overseen by a Reviewing Editor and Eve Marder as the Senior Editor. The reviewers have opted to remain anonymous.

The reviewers have discussed the reviews with one another and the Reviewing Editor has drafted this decision to help you prepare a revised submission.

The manuscript has been improved but there are some remaining issues that must be addressed before acceptance. This is an important paper that will gain from clarity in the exposition and the correction of numerous typographical errors. Of note:

1) A table of symbols and their definitions will be of great help for the reader.

2) Confusion remains in the implementation of the original clustering process as detailed by the first reviewer. Simply, a clear delineation of the process to create a template and use the template should be completely defined pictorially as well as mathematically.

3) The reviewers identify other confusing issues, missing references, and the need to avoid "double counting" by using correlation of the same data set for both analysis and verification.

Please address each of the reviewer criticisms in a written reply that addresses each reviewer comment and notes all changes to the text. The revised manuscript will be assessed by the Reviewing editor.

*Reviewer #1:*

The authors have improved the clarity of the manuscript substantially. My prior concerns have been adequately addressed. In addition to the specific comments below I would ask for additions/clarifications on three subjects.

I) First, on the path of the computation: Figure 1 is clearly the axis for most readers to understand the spike sorting process. Having carefully read the revised text as well as the original, I am still not sure how the original clustering is done. For the step of template collection, the text seems to imply that there are N clustering steps for the N electrodes. If true, one expects that each neuron would be detected on 3-8 electrodes and thus there are 3-8 times more clusters than neurons at least. The content of 1 C implies that these very many clusters are reduced to a template with 57 site "snipets" that make up the template. However, the example of 1C is centered on a maximum amplitude transient. How those maximum amplitude "centers" for templates are found is not clear to me, nor is how the expected variation around that center accommodated. Perhaps it does not need to be, since the template field is large so even is the maximum amplitude site for a given spike is off center, there is still ample information to cluster. A clear delineation of this template creation and use process would be useful, especially if it can be done pictorially as well as mathematically. How are all these redundant units merged?

II) While reading the Materials and methods section especially, but also reading the manuscript, I was constantly looking back through the text for the definition of variables and symbols. Would it be possible for the authors to make a variable definition table so that a reader can have access in one place to be reminded of variable definition for all variables in the paper?

III) The use of BEER is much better explained, but it would be useful to discuss when this assumption of elliptical boundaries is unlikely true, such as bursting cells or electrode-tissue drift.

*Reviewer #3:*

Let me begin by stating that it is very worthwhile to have a publication that describes SpyKING CIRCUS, since it is one of a handful of tools aimed at the automated analysis of multielectrode data.

Some comments follow:

1) This paper needs significant proofreading/editing to make it more readable.

As an example, the second sentence in the second section (Results) is:

“First, spikes are being detected as threshold crossings (Figure 1A) and isolated the extracellular waveforms for a number of randomly chosen spike times.”

I know what the authors intend, but it is garbled.

2) The Results section contains a description of the algorithm. Is that the format *eLife* requires? It would be clearer to have sections on the Algorithm, experimental results, validation experiments, etc.

3) A distinguishing feature of the method is the ability to resolve overlapping spikes in large systems in a reasonably automatic fashion and makes this an interesting contribution.

4) Other groups have used artificial template/spike insertion as a way to validate the algorithm and some reference could be given (e.g. Rossant 2016, Chung 2017).

5) The authors use cross-correlograms in order to help automate the merging process. It should be noted that, as a side effect, one can no longer use cross-correlograms as a validation metric.

6) Equation 1 is very confusing. The authors should explain the double sum notation (why index over events and units)? I'm sure they have something in mind but I don't know what it is.

---

## [Author Response]

[Editors’ note: the author responses to the first round of peer review follow.]

Reviewer #1:These authors report their method to do classic spike sorting, creation of single neuron spike time records, from multisite recording. They report results on both multielectrode arrays, used to record in vitro from retinal tissue, and Si probes, used to record in vivo, most frequently from rodents. Their method has been previewed on bioRxiv for some time so there is some public information comparing performance of this method to existing and other recently developed packages. While the technical treatment is thorough, I find the paper confusing, in large part because of the bifurcation of the Materials and methods and the remainder of the paper. To begin, the authors say "those snippets are detected, they are projected in a feature space of lower dimension". What feature space. Ten pages later I find out PCA. How many dimensions, either 5 or 6 depending on the parts of the algorithm. Without comment, the templates are 57 sites. This seems exceptionally large, perhaps usual for MEA recording. The choice of size deserves a clear comment.

We would like to thank the reviewer for the comment. The manuscript has been intensively rewritten, and we hope that the Materials and methods and the paper are easier to understand for the reader.

I commend the authors on their contribution of "ground truth" MEA data. The field is seriously deficient in reference data and this deficiency impairs progress in the computational problem at hand. Figure 2A and B are commendably clear. I find 2C to be puzzling. I would assume only the inset of retina in vitro is of any interest to the practitioners of neural recording. Error rates above 20% are of no practical utility but occupy 95% of the plot. Alternatively only 1 data point in the in vivo plat is useful. This result is not surprising given the traces in 2B. Peak to peak noise is ~10-12 uV. The usual peak threshold is 3.5-5 x RMS or 12-15 uV. As such, one would not expect to sort signal below that amplitude. With the authors template matching routine, it can be argued that smaller signals can be found but they are buried in noise, again as Figure 2C seems to illustrate. I am left wondering what 2C teaches, except, that sorting has failed in most of the conditions plotted. Does this not call into question the success of the strategy of finding template pools at high amplitude then matching those templates to the raw traces? If that is not the lesson, then the authors need to clarify the value of plotting so many failed sorting exercises. I fail to understand the value of a 50% classification error that matches the optimal classification rate. I would conclude the experiment has failed to produce adequate usable data.

We thank the reviewer for appreciating the value of ground truth MEA data. We agree that, in the major part of our previous recordings, the spike recorded with the loose patch electrode was too low on the MEA to be properly isolated. We have therefore collected more data where the spike height measured on the multi-electrode array is large enough so that the error rate is very low (18 cells below 10% ). We found that our spike sorting algorithm was successful for all these cells. Our algorithm is therefore able to sort accurately cells where a very low error rate as expected.

Further, the authors choose to compare their sorting accuracy to a previously defined metric, Best Ellipsoidal Error Rate. Harris, 2000, the original source defines this measure as "This measure was designed to estimate the optimal clustering performance for a given set of feature vectors by searching over all possible ellipsoidal cluster boundaries." Is not the principal weakness of the methods of Harris reported there and later an assumption that a cluster is well described by ellipsoidal boundaries? The authors should justify this assumption in the reference data sets. I see no a priori reason that their data should be so constrained, although the "ground spikes" may be.

We agree that the comparison with the BEER estimate was not properly justified. Our motivation is that all cells cannot be sorted with equal performance: if a cell evoked small spikes on the array, or if the extracellular waveforms are very similar to the ones triggered by another neighbour cell, then it will be difficult to sort this cell, and even the best algorithm will produce a large error rate. We thus need to compare the performance of our algorithm with an estimate of how well the “best” algorithm should perform. Unfortunately, there is no known method to provide a perfect estimate of this best performance. One simple option is to plot the error of our algorithm against the height of the spike. We did this and show that the error rate decreases when the spike height increases, as expected (see Figure 2C). However, this does not tell us if this is the best performance reachable. The BEER estimate is just one attempt at getting close to this best performance. We agree with the reviewer about the limitation of the BEER estimate: it assumes that spikes can be found in a cluster with ellipsoidal boundaries, which is probably wrong. In particular, spikes in overlap with spikes from another cell will probably be missed. The BEER estimate cannot be considered as estimating the optimal error rate. We used it because it has been used in several papers describing spike sorting methods (Harris et al., 2000, Rossant et al., 2016, Kadir et al., 2014) and has been established as a commonly accepted benchmark. To our knowledge, there is no alternative method that circumvents the issues of the BEER. We have mentioned the limitations of the BEER in the text. When compared to the error rate of our algorithm, we found similar results to the BEER. While this cannot be taken as a proof that our algorithm is optimally sorting the spikes, the fact that the error rate is close to a supervised classifier learned on the ground truth data suggests that it does very well.

In the Discussion, especially Figure 3, trends are difficult to understand. For 3A, the plot seems to argue a superlinear dependence on processor count, 6 processors taking 1/30th the time of 2 processors. This seems implausible. Would this plot be more informative on a log time scale without a trend-line. The current plot implies that 10 processors would digest the data in negative time! 3B is much more informative, showing 400 channels would take ~20 hours of computation for 1 hour of recording. While an improvement over many current methods, is not the method of Pachatariu et al., reference in this manuscript as a bioRxiv but now also published in Advances In Neural Information Processing Systems 4448-4456 (2016). The GPU based system reported their quotes "First, we timed the algorithm on several large scale datasets. The average run times for 32, 128 and 384 channel recordings were 10, 29 and 140 minutes respectively, on a single GPU-equipped workstation". While they did not quote the duration of the recording or the total number of neuronal event (a better metric of the computation burden) it seems far faster than the ~30 hours in the present manuscript. Both of the packages are available in the public domain so a direct comparison is a reasonable request. In Figure 3E, the authors plot data ranging from 0 to 15% (including error bars) to 50% full scale. Why not use the full plot height for 15%. Same comment for 3F.

We apologize for the confusion. The time to run the algorithm is inversely proportional to the number of CPUs. We have modified the plot to make this clearer. We have also performed a direct comparison with KiloSort (Pachitariu et al., 2016). For a small enough number of electrodes, Kilosort is faster than our algorithm. However, this comes at the cost of a much larger memory usage. As a consequence, KiloSort cannot handle large recordings with more than 500 channels, while our algorithm could handle up to 4225 channels.

Finally, the authors freely admit their strategy over segments but offer a combination of an automated merging tool as well as a "dedicated GUI" for manual merging. This reviewer did not implement that interface so I cannot comment directly on performance or ease of use. In the Materials and methods section, they describe in detail the method for correlation, but they do no show the error rate as a function of the parameters used for that merging.

We have now extensively described this tool for automated merging, and shown a solution where merging can be automated with only two parameters. We have tested this fully automated strategy using the ground truth data and found that it allows sorting neurons with a good accuracy.

In summary, I am undecided. The authors use an admirable amount to "ground truth" data, both their own and that available publicly. Unfortunately, all of these data are inadequate to the problem since the known cells are so few and in the experimental data of such low amplitude yield a residual question of performance remains. This manuscript reports a novel combination of strategies but does not appear to exceed in performance or speed other recently published packages. More than one option is probably justified based on different needs of data sets, but that optimum path is difficult to discern from this manuscript of other published data. Reporting the method is a service to the community and justified in journal space, especially for an online journal such as eLife.

We hope that the novel data and improvements in the algorithm will now convince the reviewer. The new datasets have spikes with high amplitude. Our algorithm for spike sorting also allows scaling up thousands of electrodes, which is not the case of other published software.

Reviewer #2 (General assessment and major comments (Required)):In this report, the authors present a new approach to spike sorting that they claim is scalable to thousands of electrodes. This a very timely report as there is renewed interest in spike sorting thanks to the new electrode technologies. However, I find this report to be mixed in terms of strength of the proposed solution and clarity with which the underlying claims are made.The strength of the proposed solution is significantly under-cut by the need for manual cluster merging. This is a really major limitation. Clustering of thousands of electrodes has to be automatic if it is to be genuinely scalable. I can imagine it would take a long time for someone to check thousands of recordings and merge, especially if they are not an expert.The authors cite several algorithms that are designed to scale and says they are untested. This is a fair point, but why don't they test them with their ground truth data and compare the results? This would seem to be an important issue for the reader to assess the value of the proposed work in the context of the cited literature.

We thank the reviewer for this suggestion and have now designed our own strategy to automate fully the spike sorting, with a final step of automated merging. We have tested this automated merging step with our ground truth data and found that we could reach very good performance.

The Introduction implies that the spike sorting problem has been addressed for small numbers of contacts, but does not include any citations to support the point. What forms of the classical approach that the authors are referring to? I would claim that despite a great deal of attention it is not the case that low-density sorting has been solved. The high density electrode case that the authors focus on here presents new challenges but the authors need to acknowledge the known limitations of low-density approaches.

We apologize for the confusion. We did not imply that spike sorting was solved for small numbers of contacts. However, it has received a lot of attention and several solutions have been proposed, although we agree it is not clear if these solutions are optimal or not. The common point of most of these solutions is to use clustering and we have now emphasized the limitations of a “pure” clustering approach in the Introduction and Results.

A novelty of the proposed approach is that it claims to address the superposition problem but how well this works is not demonstrated or discussed. I think the work would be strengthened if this aspect received more attention.

The superposition problem is a salient issue when trying to estimate pairwise correlations between cells. If two nearby cells spike synchronously, their spikes will temporally and spatially overlap. A clustering-based approach will miss these synchronous spikes and, as a result, the correlation between the two cells will be underestimated (see Pillow et al., 2013). To demonstrate that our algorithm could solve this problem we simulated two correlated spike trains and added these artificial cells to real extracellular recordings. We therefore generated ground truth data where we know the true correlation value for these two points. We then showed that our algorithm was able to recover the correct value for correlation. Our algorithm is therefore able to solve the superposition problem. We have added these explanations in the text.

Reviewer #3:[…] My only concerns therefore relate to the presentation of the material. Given that this is a presentation of a spike sorting resource, it seems mandatory that the computational approach and the technical aspects of the algorithm be explained as thoroughly and clearly as possible so that readers and users can fully understand the approach and adjust it to their own needs by tuning the appropriate parameters. In the Results part, the explanation of the algorithm is rather brief, and it would here be useful to provide a bit more of an intuitive description of the approach (see detailed points below). The Materials and methods section, on the other hand, appears to contain some inaccuracies and small omissions of detail that make it hard to follow some of the steps.

We have completely rewritten the Results and Materials and methods section. The results are now more complete, and the methods more accurate.

1) The Results part is a bit sparse on explaining the approach, although this would be a nice opportunity to lay out concepts without the need for technical details. For example, it might be useful to provide some brief explanations of why limiting the clustering to the search for centroids is less demanding and what the approach in density-based clustering is. Furthermore, some explanation of what is meant by "clustering each group in parallel" would be helpful, including the strategy for finding the right number of clusters and merging templates and units when necessary.

We have done our best to explain our clustering strategy here. When we search for cluster centroids, we can afford errors in the cluster borders. This is why this task can be considered as less demanding than a full sorting by clustering, where borders have to be correct. We have also detailed more what we meant by clustering in parallel: before clustering, we group spikes according the electrode where they peak. This creates as many groups as electrodes and we cluster each group independently.

2) One aspect that doesn't become clear in the main part is to what extent merging of units generally needs to be performed and how it was done for the results of Figures 2 and 3. The Materials and methods section states that several templates were used for comparison with ground-truth data. It would be useful to obtain some more information about this. Was this merging done with the GUI described in the Discussion? Was the merging done "blindly", that is, before comparison with the ground-truth data, or retrospectively? Was merging the norm or the exception?

This part has been extensively rewritten and modified following the comments of the other reviewers. We have now designed a fully automated method for the merging step, and tested it on the ground truth data.

3) A potential issue with automated spike sorting is that it might be difficult to judge the sorting quality of the obtained units, that is, how reliable the units are and to what degree one may expect contamination from other cells or missed spikes. Does the resource provide for any quality control? Useful simple statistics might be the normalized spike amplitude, the distance from other templates, the quality of the fits in the template matching phase, and the variability in the fitting parameters, but maybe the authors could also point out other ways of assessing sorting quality from their own experience of using their software.

From our ground truth data one can see that the quality of the sorted units is directly related to the size of the spike waveform. We have not found any other parameter that played a significant role in determining the quality of the sorted unit. We have added a panel showing the relation between sorting quality and spike size.

[Editors' note: the author responses to the re-review follow.]

The manuscript has been improved but there are some remaining issues that must be addressed before acceptance. This is an important paper that will gain from clarity in the exposition and the correction of numerous typographical errors. Of note:1) A table of symbols and their definitions will be of great help for the reader.2) Confusion remains in the implementation of the original clustering process as detailed by the first reviewer. Simply, a clear delineation of the process to create a template and use the template should be completely defined pictorially as well as mathematically.3) The reviewers identify other confusing issues, missing references, and the need to avoid "double counting" by using correlation of the same data set for both analysis and verification.Please address each of the reviewer criticisms in a written reply that addresses each reviewer comment and notes all changes to the text. The revised manuscript will be assessed by the Reviewing editor.

This revised version aims at addressing all of the reviewers’ criticisms. The reviewers’ concerns are addressed in detail below. We thank the reviewers for their helpful comments and criticisms, which have helped to clarify this manuscript. We have answered to all their points:

- The text has been proofread to improve clarity;

- Figure 4 has been redrawn with new colors, to improve visibility (see comment from reviewer 1);

- We added a table at the end of the main manuscript to explain all the notations used in the paper;

- We added a supplementary figure to better explain pictorially how the clustering is parallelized (see first comment of reviewer 1).

Reviewer #1:The authors have improved the clarity of the manuscript substantially. My prior concerns have been adequately addressed. In addition to the specific comments below I would ask for additions/clarifications on three subjects.I) First, on the path of the computation: Figure 1 is clearly the axis for most readers to understand the spike sorting process. Having carefully read the revised text as well as the original, I am still not sure how the original clustering is done. For the step of template collection, the text seems to imply that there are N clustering steps for the N electrodes. If true, one expects that each neuron would be detected on 3-8 electrodes and thus there are 3-8 times more clusters than neurons at least. The content of 1 C implies that these very many clusters are reduced to a template with 57 site "snipets" that make up the template. However, the example of 1C is centered on a maximum amplitude transient. How those maximum amplitude "centers" for templates are found is not clear to me, nor is how the expected variation around that center accommodated. Perhaps it does not need to be, since the template field is large so even is the maximum amplitude site for a given spike is off center, there is still ample information to cluster. A clear delineation of this template creation and use process would be useful, especially if it can be done pictorially as well as mathematically. How are all these redundant units merged?

We thank the reviewer for this comment. We would like to clarify how we do this clustering step. Even if a spike is detected on several electrodes, it will only be assigned to the electrode where the voltage peak is the largest. Thanks to this method, if a spike has its largest peak on electrode 1, but is also detected on electrode 2, it will only be assigned to electrode 1. This means that, in general, the spikes of one cell will be assigned to only one electrode and will therefore correspond to a single cluster, not to 3-8 ones as predicted by the reviewer. We have explained this better in the text, and a supplementary figure to explain this procedure has been added to the manuscript.

II) While reading the Materials and methods section especially, but also reading the manuscript, I was constantly looking back through the text for the definition of variables and symbols. Would it be possible for the authors to make a variable definition table so that a reader can have access in one place to be reminded of variable definition for all variables in the paper?

We would like to thank the reviewer for this suggestion. A table has been added at the end of the paper.

III) The use of BEER is much better explained, but it would be useful to discuss when this assumption of elliptical boundaries is unlikely true, such as bursting cells or electrode-tissue drift.

The reviewer is right, and this is now better discussed in the manuscript. Indeed, the BEER is only an approximation of the expected best performance, as this non-linear classifier may not cope perfectly with temporal overlaps, drifts, or bursting behavior. Nevertheless, in our case, since we used only few minutes long recording, the drifts appear to be negligible.

Reviewer #3:Let me begin by stating that it is very worthwhile to have a publication that describes SpyKING CIRCUS, since it is one of a handful of tools aimed at the automated analysis of multielectrode data.Some comments follow:1) This paper needs significant proofreading/editing to make it more readable.As an example, the second sentence in the second section (Results) is:“First, spikes are being detected as threshold crossings (Figure 1A) and isolated the extracellular waveforms for a number of randomly chosen spike times.”I know what the authors intend, but it is garbled.

We would like to thank the reviewer for this comment on SpyKING CIRCUS. We have proofread the manuscript, in order to make it easier to understand. We hope that this is now suitable for publication.

2) The Results section contains a description of the algorithm. Is that the format eLife requires? It would be clearer to have sections on the Algorithm, experimental results, validation experiments, etc.

Many articles in *eLife* in the Tools and Resources category include in the results a brief description of the methods, which is expanded in the Materials and methods section. For example, see

- https://elifesciences.org/articles/28728

- https://elifesciences.org/articles/32671

- https://elifesciences.org/articles/24656

We have followed this organisation. This allows to grasp the main steps of the algorithm quickly for the standard user of the algorithm, while giving all the necessary details in the Materials and methods section.

3) A distinguishing feature of the method is the ability to resolve overlapping spikes in large systems in a reasonably automatic fashion and makes this an interesting contribution.

We would like to thank the reviewer for the comment. We have emphasized more this point in the text.

4) Other groups have used artificial template/spike insertion as a way to validate the algorithm and some reference could be given (e.g. Rossant 2016, Chung 2017).

We thank the reviewer for the comment. In fact, in the original manuscript, the reference to (Rossant et al., 2016) was already there in the Materials and methods section (Simulated ground truth tests). Indeed, we are fully aware that many groups have used this strategy of designing hybrid" datasets as a way to validate spike sorting with artificial data. The reference to (Chung et al., 2017) has been added. We have also mentioned other papers where this was done previously (Segev et al., 2004, Marre et al., 2012)

5) The authors use cross-correlograms in order to help automate the merging process. It should be noted that, as a side effect, one can no longer use cross-correlograms as a validation metric.

The reviewer is right, and this is something that has now been explicitly stated in the manuscript.

6) Equation 1 is very confusing. The authors should explain the double sum notation (why index over events and units)? I'm sure they have something in mind but I don't know what it is.

We apologize for the lack of clarity. The manuscript has been modified in order to better explain this key equation. The core idea of this equation is that for every spike detected (indexed by *i*), we assume that several templates (indexed by *j*) could re and participate to this spiking event. This is why there is a double sum here. That said, since only a few cells will fire at the same spike time, most of the coefficients should be 0. This notation is nevertheless convenient to explain how spikes can be superimposed. We have clarified this in the text.